**Annual Maps of Forest Cover in the Brazilian Amazon from Analyses of PALSAR and MODIS Images**

Yuanwei Qin[1], Xiangming Xiao[1*], Hao Tang[2], Ralph Dubayah[3], Russell Doughty[4], Diyou Liu[5], Fang Liu[1], Yosio Shimabukuro[6], Egidio Arai[6], Xinxin Wang[7], Berrien Moore III[4]

[1]Department of Microbiology and Plant Biology, University of Oklahoma, Norman, OK 73019, USA

[2]Department of Geography, National University of Singapore, 1 Arts Link, Kent Ridge, Singapore 117570

[3]Department of Geographical Sciences, University of Maryland, College Park, MD, USA

[4]College of Atmospheric and Geographic Sciences, University of Oklahoma, Norman, OK, 73019, USA

[5]College of Land Science and Technology, China Agricultural University, Beijing 100083, China

[6]Brazilian National Institute for Space Research, INPE, São José dos Campos, SP, Brazil

[7]Coastal Ecosystems Research Station of the Yangtze River Estuary, Ministry of Education Key Laboratory of Biodiversity Science and Ecological Engineering, Institute of Biodiversity Science, School of Life Sciences, Fudan University, Shanghai 200438, China

*Correspondence to*: Xiangming Xiao (xiangming.xiao@ou.edu)

**Abstract.** Many forest cover maps have been generated by using optical and/or microwave images, but these forest cover maps have large area and spatial discrepancies. To date, few studies have assessed forest cover maps in terms of two biophysical parameters used in forest definition: (1) canopy height and (2) canopy coverage. We generated annual forest cover maps from 2007 to 2010 and evergreen forest cover maps from 2000 to 2021 in the Brazilian Amazon using the images from the Phased Array type L-band Synthetic Aperture Radar and the time series images from the Moderate Resolution Imaging Spectroradiometer, using the forest definition of the Food and Agriculture Organization (FAO) of the United Nations (> 5-m tree height and > 10% canopy coverage) as the reference. We used the canopy height and canopy coverage datasets from the Geoscience Laser Altimeter System during 2003-2007 to assess annual forest cover maps from 2007 to 2010 and annual evergreen forest cover maps from 2003 to 2007, and the results show high accuracy of these forest cover and evergreen forest cover maps in the Brazilian Amazon. These annual forest cover maps and annual evergreen forest cover maps provide data support for the analyses of the causes, processes, and consequences of forest cover changes in the Brazilian Amazon.

## 1. Introduction

The global forest area is $40.6 \times 10^6$ km² and accounts for ~31% of the total land area, according to the 2020 Global Forest Resources Assessment (FRA) published by the Food and Agriculture Organization (FAO) of the United Nations (FAO, 2020). Forests, especially tropical forests, play major roles in the carbon cycle (Fan et al., 2019; Mitchard, 2018), water cycle (Lovejoy and Nobre, 2019), and biodiversity (Jenkins et al., 2013). Tropical forests contain ~230 Pg C of aboveground

biomass, which is ~40% to ~60% of the carbon stored in the Earth's terrestrial vegetation (Baccini et al., 2012; Saatchi et al., 2011). However, large areas of forests have been deforested for agriculture, wood, and charcoal production (FAO, 2020). A total of $1.8 \times 10^6$ km$^2$ of forests have been lost since 1990 due to human activities and natural disturbances, although the rate of net forest loss has declined slightly (FAO, 2020). The extensive forest area loss has caused significant losses of forest carbon stock (Mitchard, 2018) and biodiversity (Ochoa-Quintero et al., 2015). Thus, updated and accurate annual maps of forests are

essential for us to track and assess the loss and gain of forest area and their impacts on carbon, water, climate, and biodiversity.

Satellite remote sensing can observe large land surface areas at regular temporal resolutions, which is suitable for generating annual maps of forests at regional, continental, and global scales. Substantial progress has been achieved in identifying and mapping the spatial and temporal changes of forests across local, regional, and global spatial scales by using various image datasets and algorithms (Souza et al., 2020; Fao, 2020; Hansen et al., 2013; Shimada et al., 2014). The time

series optical remote sensing images at coarse and moderate spatial resolutions, such as the Advanced Very High Resolution Radiometer (AVHRR, 1000-m) and MODIS (500-m and 250-m), have been widely used to generate global or continental forest cover maps, which were based on forest spectral and phenology features (Friedl et al., 2010; Hansen et al., 2002). Before 2008, optical images at a high spatial resolution (e.g., Landsat) were expensive and often used to map forests in the hotspots of deforestation, such as the Brazilian Amazon (Skole and Tucker, 1993; Inpe, 2023). In 2008, all the 30-m Landsat images

became freely available to the public (Woodcock et al., 2008). Google Earth Engine, a powerful cloud computing platform, was also developed to process these big datasets (Gorelick et al., 2017). Continental or global tree cover and forest cover maps at a high spatial resolution have been generated using Landsat images (Sexton et al., 2015; Hansen et al., 2013; Souza et al., 2020).

In comparison to the optical images, images from microwave sensors, in particular, L-band synthetic aperture radar

(SAR), have two advantages: (1) they are affected less by clouds and cloud shadows, and (2) they have stronger penetration capability into forest canopy and interact with tree branches and trunks, and thus, are more sensitive to forest structure and aboveground biomass. SAR images have advantages in forest mapping, especially in the tropical region where clouds and cloud shadows are frequent (Qin et al., 2017; Shimada et al., 2014; Qin et al., 2016a; Chen et al., 2018; Reiche et al., 2016). In the Advanced Land Observing Satellite (ALOS), the Phased Array type L-band Synthetic Aperture Radar (PALSAR) is the

first L-band microwave remote sensing sensor to carry out global land surface observations (Shimada et al., 2014). The 25-m and 50-m ALOS PALSAR data have been used to map regional and global forest and forest change (Shimada et al., 2014; Qin et al., 2017; Chen et al., 2018; Thapa et al., 2014). However, some non-vegetation land cover types, such as rocky land, bare land, and buildings, have high backscatter signals similar to those of forests, and they are misclassified as forest, resulting in commission errors in the forest cover maps. The combination of microwave and optical images allows us to produce improved

forest cover maps with reduced commission errors. Thus, we developed a simple but robust algorithm to identify and generate annual forest cover maps using both ALOS PALSAR and MODIS images, and the algorithm has been successfully applied to map forest cover in Asia (Qin et al., 2015; Qin et al., 2016a) and South America (Qin et al., 2017; Qin et al., 2019).

Previous studies suggested that forest definition is one of the major reasons for the discrepancy of forest cover maps (Sexton et al., 2015), and our study showed the validation data affect the accuracy assessment of forest cover maps, although these forest cover maps had the same forest definition (Qin et al., 2017). Accuracy assessments and uncertainty analyses of the forest cover maps have been carried out using reference datasets from various approaches, including field surveys, images with higher spatial resolutions, and previously available land cover maps (Xiao et al., 2011; Fritz et al., 2012; Olofsson et al., 2012; Stehman et al., 2012; Tyukavina et al., 2017). Field surveys are carried out either over a region or in situ that aim to track long-term forest area and biomass changes under human activities and natural disturbances (Matricardi et al., 2020). These field forest surveys are time-consuming and have a high labor cost; thus, limited field samples are collected, which may introduce large bias and are not suitable to assess the forest cover maps at continental or global scales (Tang et al., 2019a). The higher spatial resolution images and land cover maps do not have accurate canopy height information, which is one of the primary criteria for forest definition.

Canopy height and canopy coverage are two important characteristics of forests. Airborne lidar observations are being used to accurately measure canopy height and canopy coverage, but these measurements are mainly carried out at a local scale (Tang et al., 2019b; Hudak et al., 2002; Leitold et al., 2018). The satellite Lidar-based canopy height and canopy coverage monitoring has evolved remarkably, which provides an opportunity to use them to evaluate the accuracy and uncertainty of forest cover maps and improve forest cover mapping. Recently, new and reliable canopy height and canopy cover percentage datasets (Tang et al., 2019a) were generated using the Geoscience Laser Altimeter System (GLAS) onboard the Ice, Clouds, and Land Elevation Satellite (ICESat) satellite. These new datasets provide a unique opportunity to assess annual forest cover maps generated according to the FAO FRA forest definition (trees > 5-m height and 10% canopy coverage). The recently developed ICESat/GLAS canopy height and canopy coverage dataset has four features (Tang et al., 2019a). First, at the footprint level, the ICESat/GLAS canopy coverage is consistent with the airborne Lidar estimated with almost no bias. Second, ICESat/GLAS showed higher sensitivity to high canopy coverage in densely forested areas than that of optical remote sensing images. Third, ICESat/GLAS shows a stronger ability to differentiate subtle temporal changes in canopy coverage compared to optical remote sensing. Fourth, ICESat/GLAS provides unique information about the vertical canopy cover and structure, an important variable in defining tree canopy height.

Tropical forests in the Brazilian Amazon are influenced by mining, deforestation, fires, severe droughts, wind storms, and other disturbances (Tyukavina et al., 2017; Espírito-Santo et al., 2014; Aragão et al., 2018; Sonter et al., 2017; Li et al., 2019). Because of frequent clouds, cloud shadows, high aerosols, and limited accessibility of the Amazon, in situ reference data, field surveys, and very high spatial resolution images are very limited, which prevents a robust inter-comparison among these data sources. Here, we selected the Brazilian Amazon as a hotspot with a large forest area and extensive forest change. We assessed the annual PALSAR/MODIS forest cover maps at 50-m spatial resolution during 2007-2010 (Qin et al., 2017) and the MODIS evergreen forest cover maps during 2003-2007 at 500-m spatial resolution (Qin et al., 2019) using the ICESat/GLAS canopy height and canopy coverage data. To our limited knowledge, this is the first study using a large lidar-

based canopy height and canopy coverage dataset to assess the accuracy and uncertainty of annual forest cover maps in the Brazilian Amazon.

## 2. Methods

### 2.1. Study area

The Brazilian Amazon covers an area between 18˚S to 6˚N and 74˚ W to 41˚ W, and includes nine states (Acre, Amazonas, Amapá, Pará, Mato Grosso, Maranhão, Rondônia, Roraima, and Tocantins). The Brazilian Amazon has the largest tropical forests and most diverse terrestrial ecosystems in the world (Jenkins et al., 2013). Annual precipitation increases from ~1500 mm per year in the southeast to > 3000 mm per year in the northwest in normal years (Qin et al., 2019). On average, the Brazilian Amazon has an annual precipitation of ~2000 mm per year and an annual mean temperature of 27 ˚C (Almeida et al., 2017). The two major biomes in Brazil are the Amazon in the north and west, and the Cerrado, *i.e.*, a vast ecoregion of tropical savanna, in the south and east. Rapidly changing land use, disturbances (e.g., fire), climate, and other human activities have resulted in substantial deforestation and degradation in the Brazilian Amazon over the past decades (Fearnside, 2005; Nepstad et al., 2014; Matricardi et al., 2020). The rapid expansion of cropland and pasture areas makes Brazil a leading global exporter of agricultural and livestock commodities, especially soybean and beef.

### 2.2. ALOS PALSAR mosaic data and pre-processing

The ALOS PALSAR is an L-band active microwave remote sensing sensor, and PALSAR images are less affected by cloud and atmospheric conditions than optical images. The 50-m ALOS PALSAR mosaic data, provided by the Earth Observation Research Center, Japan Aerospace Exploration Agency (JAXA), include HH gamma-naught and HV gamma-naught, and three other layers (mask information, local incidence angle, and total dates from the ALOS launch date) (Shimada et al., 2014). Three major pre-processing tasks were carried out to reduce the noise in the ALOS PALSAR data (Shimada et al., 2014). The ALOS PALSAR strip data, with the minimum response to surface moisture, were used to generate the annual ALOS PALSAR mosaic data. The raw images were calibrated based on published coefficients, and outputs with 16-looks were further provided to reduce speckle noise. PALSAR HH and HV backscatter data were orthorectified and slope corrected using the Digital Elevation Model (90-m) from the Shuttle Radar Topography Mission. The Digital Number (DN, amplitude values) of HH and HV were converted to backscattering coefficients in decibels (gamma-naught, $\gamma^{\circ}$, see Equation 1).

$$\gamma^{\circ} = 10 \times log_{10} < DN^2 > + CF \quad (1)$$

Where CF is the absolute calibration factor with a value of -83 (Shimada et al., 2009). We also calculated the Difference (see Equation 2) and Ratio (see Equation 3) between HH and HV:

$$Difference = HH - HV \quad (2)$$

$$Ratio = {HH}/{HV} \quad (3)$$

## 2.3. MODIS surface reflectance and vegetation indices

The MOD09A1 (Collection 006) data product provides land surface reflectance at a spatial resolution of 500 m (463 m) after atmospheric correction, including gases, aerosols, and Rayleigh scattering. The MOD09A1 data product has seven spectral bands: Blue (459 - 479 nm), Green (545 - 565 nm), Red (620 - 670 nm), Near-Infrared (NIR, 841 - 876 nm and 1230 - 1250 nm), Shortwave Infrared (SWIR, 1628 - 1652 nm and 2105 - 2155 nm), and two quality layers. For each 8-day composite, the best-quality value is selected from all the acquisitions within every eight days. Although the 8-day MOD09A1 data product has a spatial resolution of 500m, it is generated based on daily observations and has a high opportunity to get cloud-free observations. Besides, MOD09A1 has a data collection since 2000, which could track long-term forest cover changes due to frequent policy and environmental changes in the Brazilian Amazon, especially the different phases of deforestation. We identified and excluded all observations in each image covered by cloud (cloud, internal cloud, and high cirrus), cloud shadow, high aerosols, or snow labelled in the data quality layers. We also identified and excluded all the observations with a blue band value larger than 0.2 as an additional cloud flag. We then calculated three vegetation indices for each observation in time series (see Equation 4, 5, 6): Normalized Difference Vegetation Index (NDVI), Enhanced Vegetation Index (EVI), and Land Surface Water Index (LSWI) (Xiao et al., 2002).

$$NDVI = \frac{NIR - RED}{NIR + RED} \tag{4}$$

$$EVI = 2.5 \times \frac{NIR - RED}{NIR + 6 \times RED - 7.5 \times BLUE + 1} \tag{5}$$

$$LSWI = \frac{NIR - SWIR}{NIR + SWIR} \tag{6}$$

where BLUE, RED, NIR, and SWIR represent land surface reflectance values for Blue, Red, Near Infrared (841 - 876 nm), and Shortwave Infrared (1628 - 1652 nm) bands from MOD09A1, respectively.

The MODIS/Terra Vegetation Index data product (MOD13Q1) has a spatial resolution of 250-m and a temporal resolution of 16 days. The MOD13Q1 dataset includes two vegetation index layers (NDVI and EVI). The highest NDVI and EVI values were chosen from the best daily observations every 16 days. Although the MOD13Q1 data product has a spatial resolution of 250m, its daily revisit cycle provided more opportunities for cloud-free observations in the Brazilian Amazon compared to the high spatial resolution images, such as the 30-m 16-day Landsat images. We used the observations with the property of "VI produced with good quality" based on the MOD13Q1 quality band (DetailedQA) in time series analysis. In this study, we used the NDVI layer and calculated the maximum NDVI values (NDVImax) in a year for each pixel during 2007-2010. To match the 50-m PALSAR data, we resampled the 250-m MODIS NDVImax into 50-m spatial resolution using the nearest sampling approach.

## 2.4. ICESat canopy height and cover percentage data

The ICESat GLAS, launched in 2003, is the first lidar sensor for global land surface observations. The GLAS sensor recorded the land surface elevations through time at ~65-m footprints, which have an along-track distance of about 175 m and a maximum between-track distance of about 30 km at the equator. The GLAS sensor had a revisit cycle of 91 days. We recently developed a new approach and calculated the canopy height (meters) and canopy coverage (%) datasets using the GLA14 lidar

datasets during 2003-2007 (Tang et al., 2019a). At each footprint, maximum canopy height and canopy coverage are calculated from lidar waveform signals and screened from several confounding factors (e.g., cloud, noise, and topographic slope) (Tang et al., 2019a). The derived canopy coverage showed almost no bias compared to airborne lidar estimates and was sensitive to signal dynamics over dense forests, even with canopy cover exceeding 80%. The ICESat/GLAS-based canopy height and canopy coverage estimates could better characterize footprint-level canopy conditions than the existing data products derived from conventional optical remote sensing (Tang et al., 2019a). Yet, it cannot be used to generate a wall-to-wall map of forest structure due to its limited spatial samplings.

In the Brazilian Amazon, there were about $1.1 \times 10^6$ footprints of canopy height and canopy coverage retrieved from the ICESat/GLAS observations during 2003-2007 (Fig. 1). Among these footprints, about $1.0 \times 10^6$ footprints had a canopy height of more than 5 m, which accounted for about 96% of all the footprints (Fig. 1c), and about $0.94 \times 10^6$ (86.7%) footprints had a canopy coverage of more than 10% (Fig. 1d). When considering both canopy height and canopy coverage, $0.9 \times 10^6$ footprints (85.1%) had a canopy height of $> 5$ m and canopy coverage of $> 10\%$, and these footprints were thus identified as forest footprints in terms of the FAO FRA forest definition. About 27.7% of the forest footprints had canopy height larger than 30 m and canopy coverage larger than 80%, which suggested that the Brazilian Amazon is an area with largely tall and dense trees. The ICESat/GLAS canopy height and canopy coverage data had distinct spatial distributions (Fig. 1a, b). Those tall and dense forests were mainly located in the north and west of the Brazilian Amazon, where human activities and natural disturbances are limited. Short and open forests were mainly located in the Cerrado area, south and east of the Brazilian Amazon, where extensive agriculture production occurs.

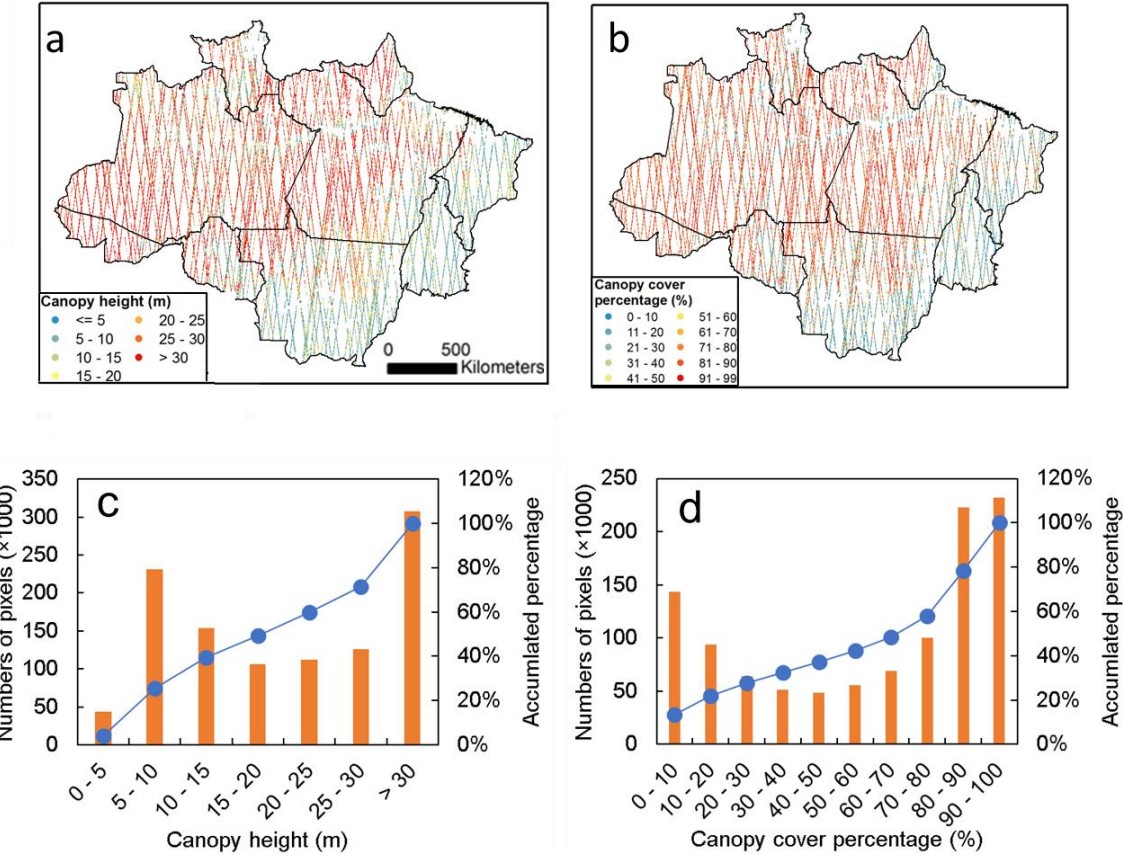

**Fig. 1.** Spatial distribution maps of ICESat-GLAS canopy height (a) and canopy coverage (b) at the footprint scale and their histograms (c-d) in the Brazilian Amazon from 2003 to 2007.

## 2.5. FAO forest definition

Hundreds of different forest definitions have been used in forest management (FAO, 2002; Lund, 2014). FAO defines forest as land with a minimum area of 0.5 hectares with (1) a tree canopy height of > 5 m and (2) a canopy coverage of >10% at the time of observations, and it also includes lands with trees that can reach these thresholds at the time of tree mature (FRA, 2020). Using the FAO's forest definition as the reference (a tree canopy height of > 5 m and a canopy coverage of >10%), we identified and generated annual PALSAR/MODIS forest cover maps in the Brazilian Amazon during 2007-2010. We defined evergreen forests as forests with green leaves year-round, a tree canopy cover more than 10%, and a tree height larger than 5 m. Then we generated annual MODIS evergreen forest cover maps from 2000 to 2021. Note that as we use satellite images to identify and map forests, we do not consider lands with trees that can reach these two thresholds at the time of tree mature. Due to various spatial resolutions of satellite images, tree distribution patterns, and terrains, the minimum forest mapping area may not be exactly 0.5 hectares.

## 2.6. Algorithm and data of annual PALSAR/MODIS forest cover maps during 2007-2010

Electromagnetic wave of PALSAR can penetrate the tree canopy and interact with the tree trunks and branches. Forests have higher volume backscatter signals in HH and HV compared to croplands, grasslands, and water bodies. Thus, PALSAR data are sensitive to forest structure and biomass. However, PALSAR data can be affected by local incidence angle and soil moisture as PALSAR data is acquired at a different date each year. We calculated the acquisition date (Fig. 2), the local incidence angle (Fig. S1), and HH and HV gamma-naught values for each year and their standard deviations (Fig. S2) during 2007-2010 in the Brazilian Amazon. PALSAR HH and HV data were mainly acquired in the dry season (from June to October) and the local incidence angle is stable. About 90% of the area has standard deviation values of less than 1 dB for PALSAR HH and HV data. PALSAR data have advantages in identifying and mapping the spatial and temporal changes of forests in the tropics with frequent clouds compared to optical satellite remote sensing. Using the FAO's forest definition as the reference, we developed a robust decision tree algorithm to identify and generate forest cover maps by ALOS PALSAR data: $-15 \leq HV \leq -9$, $3 \leq Difference \leq 7$, and $0.35 \leq Ratio \leq 0.75$, based on the forest and non-forest training samples (Qin et al., 2016a; Qin et al., 2015; Qin et al., 2017). Several land cover types (e.g., rocks and buildings) had high backscatter values of HH and HV, which were often confused with the forests when only HH and HV data were used. These land cover types usually have low vegetation coverage with $NDVImax < 0.5$ (Qin et al., 2016a; Qin et al., 2015; Qin et al., 2017). To reduce the commission errors from these land cover types, we combined both PALSAR and NDVImax from MOD13Q1 to produce annual forest cover maps (namely PALSAR/MODIS forest cover maps) at 50-m spatial resolution in the Brazilian Amazon during 2007-2010 using these threshold values: $-15 \leq HV \leq -9$, $3 \leq Difference \leq 7$, $0.35 \leq Ratio \leq 0.75$, and $NDVImax \geq 0.5$ (Qin et al., 2016a; Qin et al., 2017). We also carried out a three-year temporal consistency filter to reduce the effects of noise (Qin et al., 2016a; Qin et al., 2017).

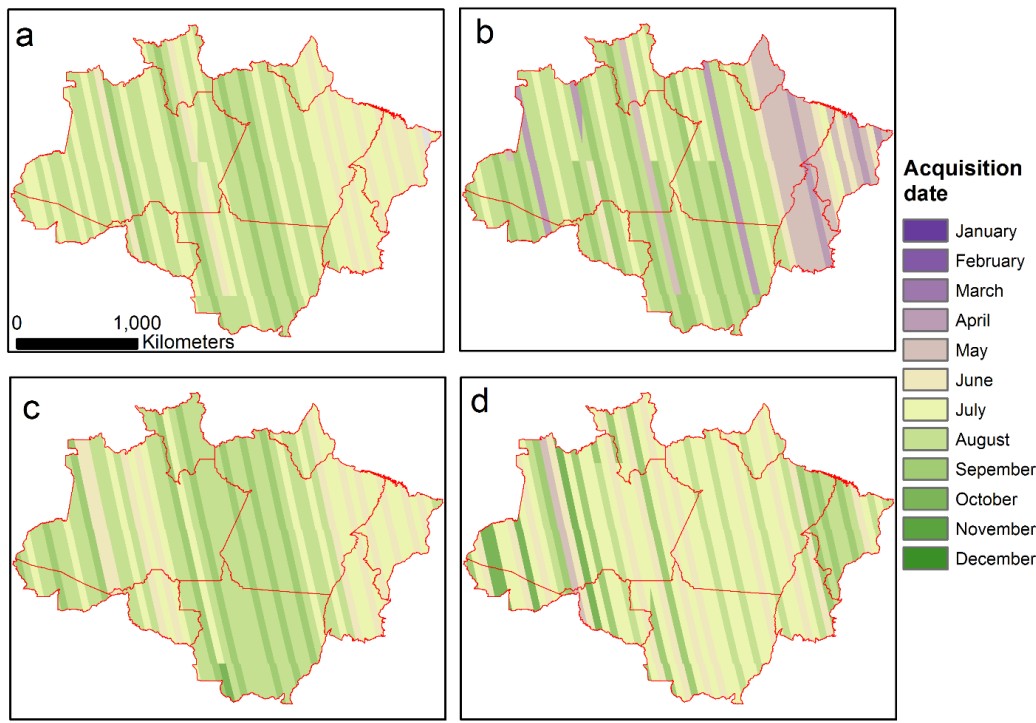

**Fig. 2.** Spatial distribution maps of ALOS PALSAR acquisition date during 2007 (a), 2008 (b), 2009 (c), and 2010 (d).


### 2.7. Algorithm and data of annual MODIS evergreen forest cover maps during 2000-2021

Evergreen forest is the dominant forest cover type in the Brazilian Amazon (Fanin and Van Der Werf, 2015). Evergreen forests have a unique biophysical feature that evergreen forests have green leaves year-round. Based on the canopy phenology from analyses of time series water-related LSWI and greenness-related EVI calculated from all MOD09A1 data in

each year, a novel, simple and robust algorithm was developed to generate annual maps of evergreen forests in the Brazilian Amazon using the FAO's forest definition as the reference and evergreen forest training samples (Xiao et al., 2009; Qin et al., 2019). We calculated (1) the frequency (percentage) of the number of observations with LSWI ≥ 0 over all the available observations ($FQ_{LSWI≥0}$) and (2) the minimum EVI values ( $EVI_{min}$) in a year after excluding observations of clouds, cloud shadows, and snow based on the MOD09A1 quality band (StateQA). We applied the Forest-MODIS algorithm ($FQ_{LSWI≥0}$ =

100% & $EVI_{min}$ ≥ 0.2) (Xiao et al., 2009; Qin et al., 2019) to time series LSWI and EVI data in a year and generated annual maps of evergreen forest from 2000 to 2021 in the Brazilian Amazon. Small numbers of MODIS pixels in the Brazilian Amazon were contaminated by clouds or aerosols, causing a significant drop in EVI values and no significant change in LSWI and NDVI values. However, these pixels may not be detected by the quality layer. Thus, we conducted an additional criterion and made a minor improvement in evergreen forest mapping, *i.e.*, $FQ_{LSWI≥0}$ > 90% & $EVI_{min}$ ≥ 0.2 & $LSWI_{min}$ ≥ 0. We also

applied a three-year temporal consistency filter to reduce the effects of noises on forest cover mapping. The Google Earth Engine (GEE) cloud platform was used for MODIS images processing.

## 2.8. Spatial and statistical analysis

We overlaid all the ICESat/GLAS footprints to the annual PALSAR/MODIS forest cover maps during 2007-2010 (50-m spatial resolution) and the annual MODIS evergreen forest cover maps during 2003-2007 (500-m spatial resolution)
and generated a table that records individual ICESat/GLAS footprint IDs, canopy height, canopy coverage, forest and non-forest. We evaluated the accuracy of these annual PALSAR/MODSI forest cover maps and annual MODIS evergreen forest cover maps in terms of canopy height and canopy coverage for all forest pixels ($1.5 \times 10^9$ pixels for PALSAR/MODIS forest and $17.5 \times 10^6$ pixels for MODIS evergreen forest each year) that contained the information of ICESat/GLAS footprint data.

For the spatial comparison, to avoid the bias caused by different spatial resolutions, we aggregated the 50-m annual
PALSAR/MODIS forest cover maps and 500-m (463-m) MODIS evergreen forest cover maps into 5-km pixels and calculated their average forest area fraction values within individual 5-km pixels. We then compared the spatial consistency between the annual PALSAR/MODIS forest cover maps and the annual MODIS evergreen forest cover maps during 2007-2010 in the Brazilian Amazon at 5-km spatial resolution based on the linear relationships and significance analyses.

For the forest area comparison, we compared the annual PALSAR/MODIS forest area and annual MODIS evergreen
forest area with multiple forest area datasets, including the PALSAR-based forest cover maps developed by JAXA (Shimada et al., 2014), the Landsat-based Global Forest Watch (GFW) dataset (Hansen et al., 2013), and the PRODES forest cover maps (INPE, 2023). We also calculated the Root Mean Square Error (RMSE) of forest areas between different forest cover data products in the Brazilian Amazon.

## 3. Results and Discussions

### 3.1. Annual PALSAR/MODIS forest cover maps during 2007-2010

Forest areas estimated by the 50-m PALSAR/MODIS forest cover maps (Fig. 3) had a small net loss in the Brazilian Amazon, decreasing from $3.77 \times 10^6$ km$^2$ in 2007 to $3.75 \times 10^6$ km$^2$ in 2010 under the influence of changing land-use policies and natural disturbances (Nepstad et al., 2014). To assess the accuracy of the PALSAR/MODIS forest cover map, we used
two independent reference datasets. First, we used the land cover maps at the 2-m spatial resolution from the Global Land Cover Validation Reference Dataset in 2010, which had land cover maps in 18 blocks ($0.15 \times 10^6$ pixels) in the Brazilian Amazon and each block covered an area of $5 \times 5$ km$^2$ (Olofsson et al., 2012; Stehman et al., 2012). Second, we used the land cover maps from the TREES-3 (Achard et al., 2014) reference dataset at the 30-m spatial resolution from the European Commission's Joint Research Centre (JRC), which had land cover maps in 416 blocks ($17.09 \times 10^6$ pixels) and each block
covered an area of $10 \times 10$ km$^2$. The Global Land Cover Validation Reference Dataset was produced from very high spatial resolution commercial remote sensing data acquired around 2010 and is freely available at the https://web.archive.org/web/20161209205946/https:/landcover.usgs.gov/glc/SitesDescriptionAndDownloads.php. The TREES-3 dataset was produced from Landsat images and is freely available at https://forobs.jrc.ec.europa.eu/trees3/data. We

calculated the error matrices (Table S1) and the overall accuracy of the PALSAR/MODIS forest cover map in 2010 was about
91% (Qin et al., 2019).

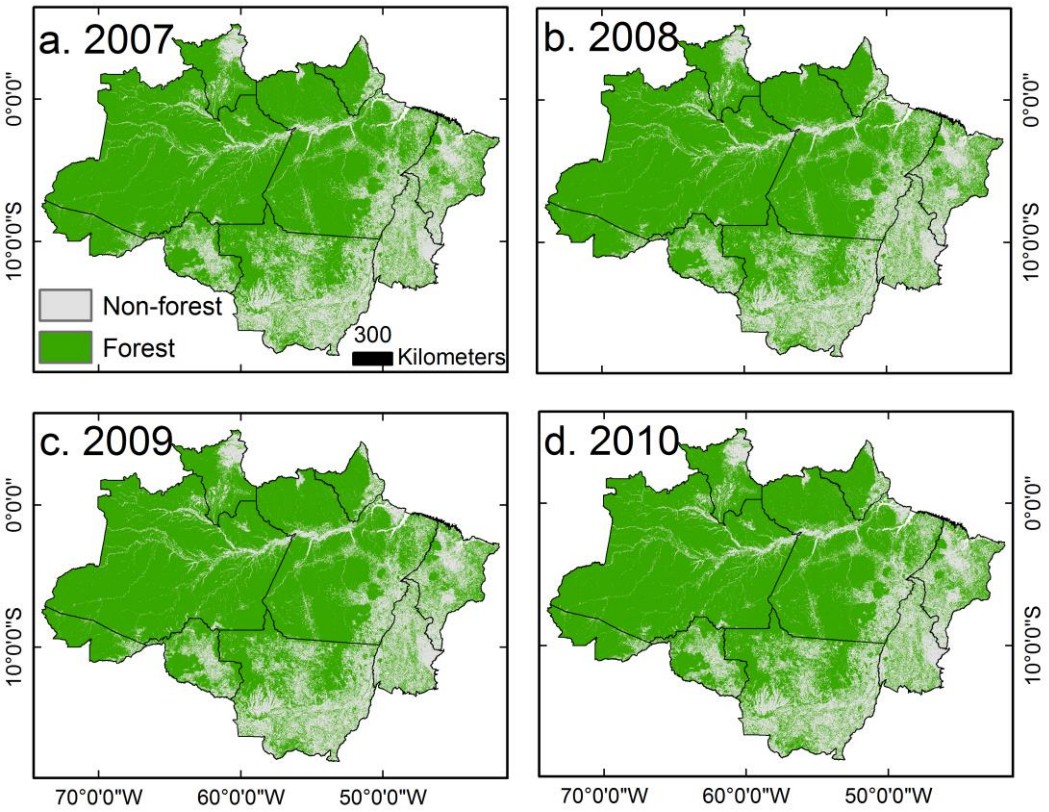

**Fig. 3.** Spatial distribution maps of annual PALSAR/MODIS forest cover maps during 2007 (a), 2008 (b), 2009 (c), and 2010 (d).

275          We used the ICESat/GLAS canopy height and canopy coverage to evaluate the accuracy of annual PALSAR/MODIS forest cover maps during 2007-2010 in the Brazilian Amazon. Fig. 4 shows the distribution of forest pixels from the annual PALSAR/MODIS forest cover maps in the 2-dimension space of canopy height and canopy coverage. About 98.5% of PALSAR/MODIS forest pixels in 2010 had canopy height > 5 meters and about 94.4% of PALSAR/MODIS forest pixels in 2010 had canopy coverage > 10%, respectively (Fig. 4). When considering both canopy height and canopy coverage, about
93.8% of PALSAR/MODIS forest pixels in 2010 had both canopy height of > 5 m and canopy coverage of > 10%. The PALSAR/MODIS forest cover maps during 2007-2009 had similar high accuracies as that in 2010 (Fig. 4). Overall, the PALSAR/MODIS forest cover maps showed high accuracies using different validation datasets.

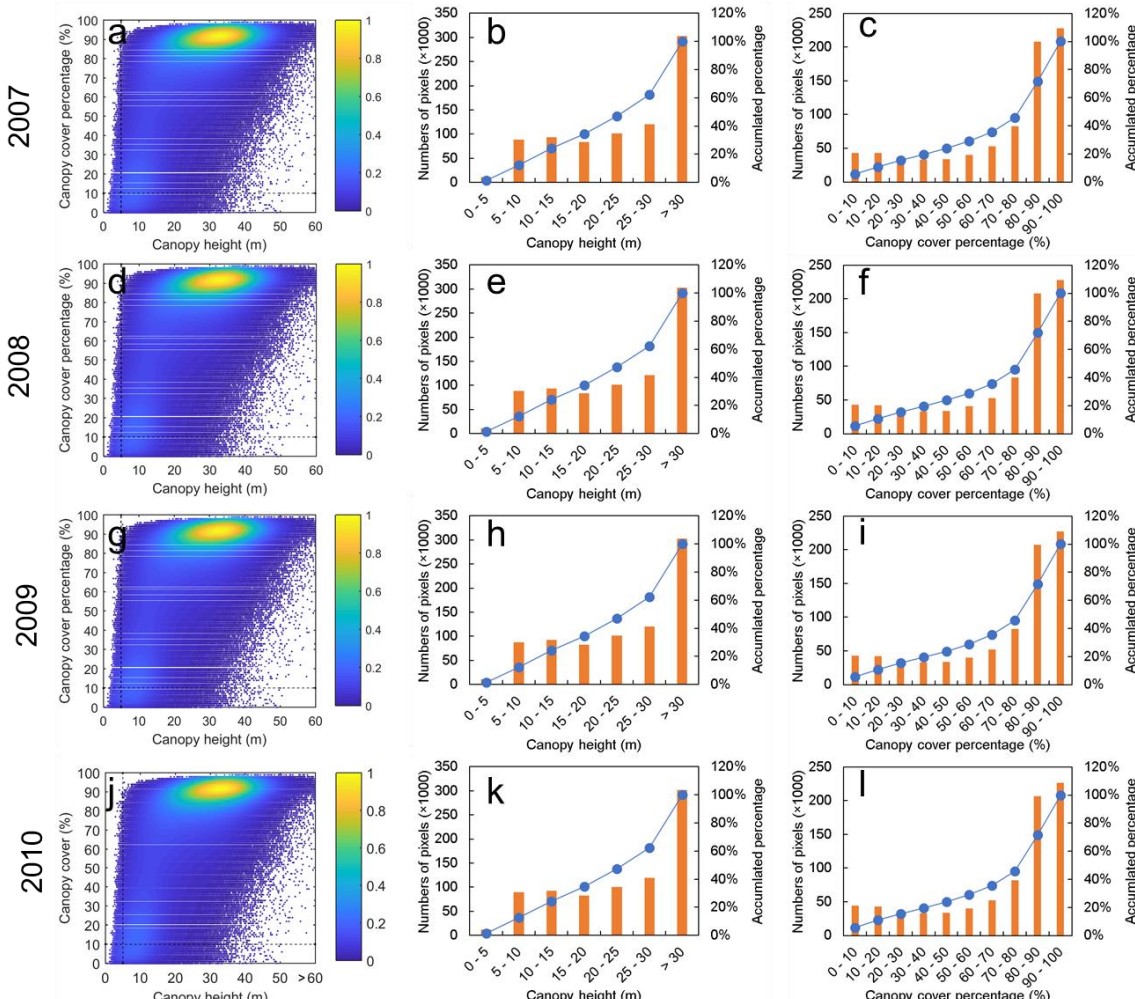

**Fig. 4.** Two-dimension scatter plots and histograms of ICESat/GLAS canopy height (m) and canopy coverage (%) at the footprint scale for the forest pixels in the annual PALSAR/MODIS forest cover maps during 2007-2010. Two-dimension scatter plots of ICESat/GLAS canopy height and canopy coverage for the forest pixels in the annual PALSAR/MODIS forest cover maps during 2007 (a), 2008 (d), 2009 (g), and 2010 (j). Histograms of ICESat/GLAS canopy height for the forest pixels in the annual PALSAR/MODIS forest cover maps during 2007 (b), 2008 (e), 2009 (h), and 2010 (k). Histograms of ICESat/GLAS canopy coverage for the forest pixels in the annual PALSAR/MODIS forest cover maps during 2007 (c), 2008 (f), 2009 (i), and 2010 (l).

Geographically, forests were distributed mainly in the Brazilian Amazon's north and west. The mixed landscapes of forest and non-forest were in the south and east of the Brazilian Amazon (Fig. 3). The State of Amazonas ($1.46 \times 10^6$ km$^2$),

Pará ($0.99 \times 10^6$ km$^2$), and Mato Grosso ($0.46 \times 10^6$ km$^2$) had the largest forest areas, which accounted for about 78% of the

total forest area in the Brazilian Amazon in 2010. The other six states had a total forest area of $0.81 \times 10^6$ km$^2$ in 2010.

### 3.2. Annual MODIS evergreen forest cover maps during 2000-2021

Evergreen forest areas also slightly decreased from $3.73 \times 10^6$ km$^2$ in 2007 to $3.72 \times 10^6$ km$^2$ in 2010 in the Brazilian Amazon (Fig. 5), consistent with the PALSAR/MODIS forest area estimated with a Root Mean Square Error (RMSE) of $0.03 \times 10^6$ km$^2$. The evergreen forest area was $3.94 \times 10^6$ km$^2$ in 2000 and then declined substantially to $3.66 \times 10^6$ km$^2$ in 2021

(Fig. 5), with a net forest area loss of $0.28 \times 10^6$ km$^2$ (7%). We previously assessed the accuracy of the evergreen forest cover map in 2010 using 18 blocks with $5 \times 5$ km$^2$ ($1.27 \times 10^3$ pixels) of land cover maps at 2-m spatial resolution from the Global Land Cover Validation Reference Dataset in 2010 and 416 blocks with $10 \times 10$ km$^2$ ($0.13 \times 10^6$ pixels) of land cover maps from the 30-m TREES-3 (Achard et al., 2014) reference dataset. The overall accuracy of the MODIS evergreen forest cover map in 2010 was about 97% (Qin et al., 2019).

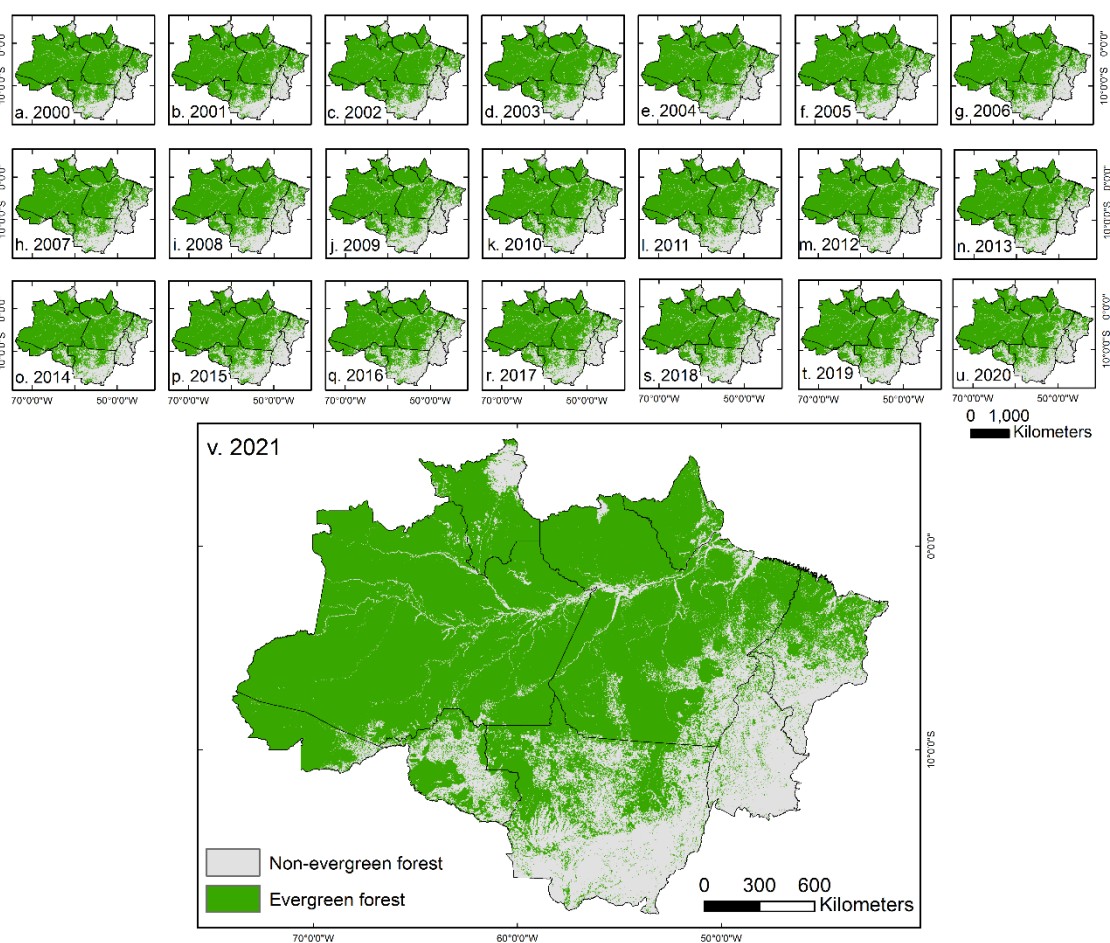


**Fig. 5.** Spatial distribution maps of annual MODIS evergreen forest cover maps in the Brazilian Amazon during 2000-2021. (a-u) Annual evergreen forest cover maps from 2000 to 2020. (v) Annual evergreen forest cover map in 2021.

Here, we used the ICESat/GLAS canopy height and canopy coverage data to evaluate the accuracy of the evergreen forest cover maps during 2003-2007 in the Brazilian Amazon. About 98.1% ($0.77 \times 10^6$ pixels) of evergreen forest pixels had canopy height of > 5 meters, and about 93.8% ($0.73 \times 10^6$ pixels) of evergreen forest pixels had canopy coverage of > 10% in 2003-2007. When considering both canopy height and canopy coverage, about 93.0% ($0.73 \times 10^6$ pixels) of evergreen forest pixels during 2003-2007 had canopy height of > 5 m and canopy coverage of >10% (Fig. 6).

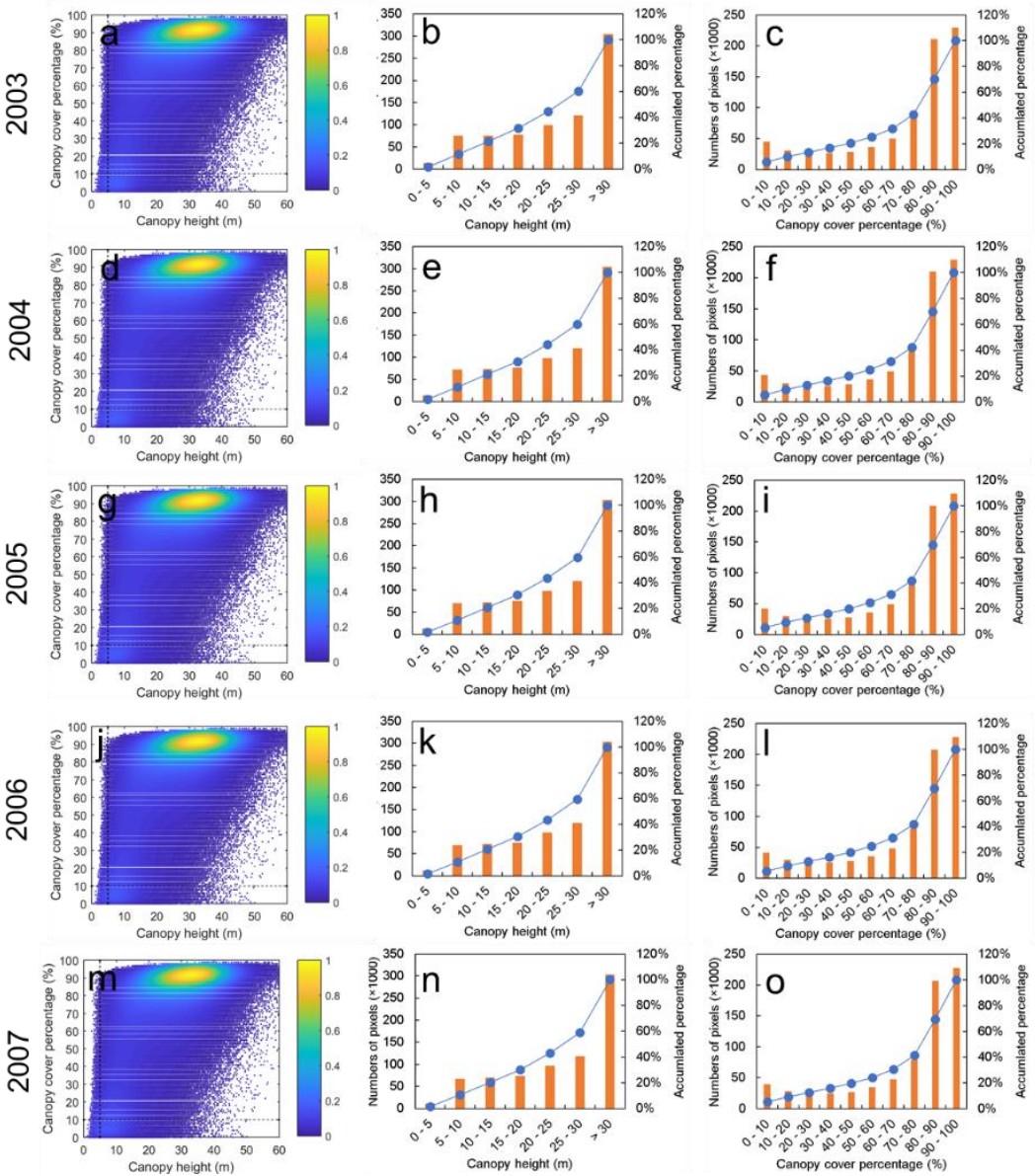

**Fig. 6.** Two-dimension scatter plots and histograms of ICESat/GLAS canopy height (m) and canopy coverage (%) for the forest pixels from the annual MODIS evergreen forest maps during 2003-2007. Two-dimension scatter plots of ICESat/GLAS canopy height and canopy coverage for the forest pixels from the annual MODIS evergreen forest maps during 2003 (a), 2004

(d), 2005 (g), 2006 (j), and 2007 (m). Histograms of ICESat/GLAS canopy height for the forest pixels from the annual MODIS evergreen forest maps during 2003 (b), 2004 (e), 2005 (h), 2006 (k), and 2007 (n). Histograms of ICESat/GLAS canopy coverage for the forest pixels from the annual MODIS evergreen forest maps during 2003 (c), 2004 (f), 2005 (i), 2006 (l), and 2007 (o).

### 3.3. Uncertainties in the accuracy assessment of PALSAR/MODIS forest cover maps and MODIS evergreen forest cover maps

The Global Land Cover Validation Reference Dataset and TREES-3 land cover dataset were produced from optical remote sensing images, which were sensitive to canopy coverage instead of canopy height. Thus, our PALSAR/MODIS forest cover maps had high user's and producer's accuracy for forest cover type, while the non-forest cover type had relatively low user's or producer's accuracy (Table S1), which may be attributed to the uncertainties in the reference maps. Unlike the optical remote sensing image, the ICESat-1 data used in this study had the maximum canopy coverage data and the maximum canopy height from 2003 to 2007 (Tang et al., 2019a). As the Brazilian Amazon had high annual primary forest loss rates of 17,654 km$^2$/yr in the 2000s (INPE, 2023), the maximum canopy height and canopy coverage of ICESat-1 data may not include the impacts of deforestation. Thus, ~94% of PALSAR/MODIS forest cover pixels and MODIS evergreen forest cover pixels meet the forest definition.

### 3.4. Spatial consistency between annual PALSAR/MODIS forest cover maps and annual MODIS evergreen forest cover maps during 2007-2010

At the regional scale, we analyzed the reasonability of the MODIS evergreen forest cover maps in the Brazilian Amazon using the PALSAR/MODIS forest cover maps as the reference maps. At the 5-km spatial resolution, the annual PALSAR/MODIS forest cover maps had a good spatial consistency with evergreen forest cover maps, especially for the dense forest during 2007-2010 (Fig. 7). The PALSAR/MODIS forest cover maps had more forest area than that of MODIS evergreen forest cover maps in the Cerrado area, south and east of the Brazilian Amazon. There might be some non-evergreen forest and sparse forest in the Cerrado area, which are not identified as forest in the MODIS evergreen forest cover maps. Overall, the forest area fraction between the PALSAR/MODIS forest cover maps and the MODIS evergreen forest cover maps was near the 1:1 linear relationship at the 5-km spatial resolution (Fig. 7 i-l), showing that the combination of PALSAR and MODIS images, and dense time series MODIS images can both generate high-accuracy forest cover maps in the Brazilian Amazon. At the site level, PALSAR/MODIS forest cover and MODIS evergreen forest cover maps showed good consistency with the very high spatial resolution images in the Brazilian Amazon (Fig. 8)

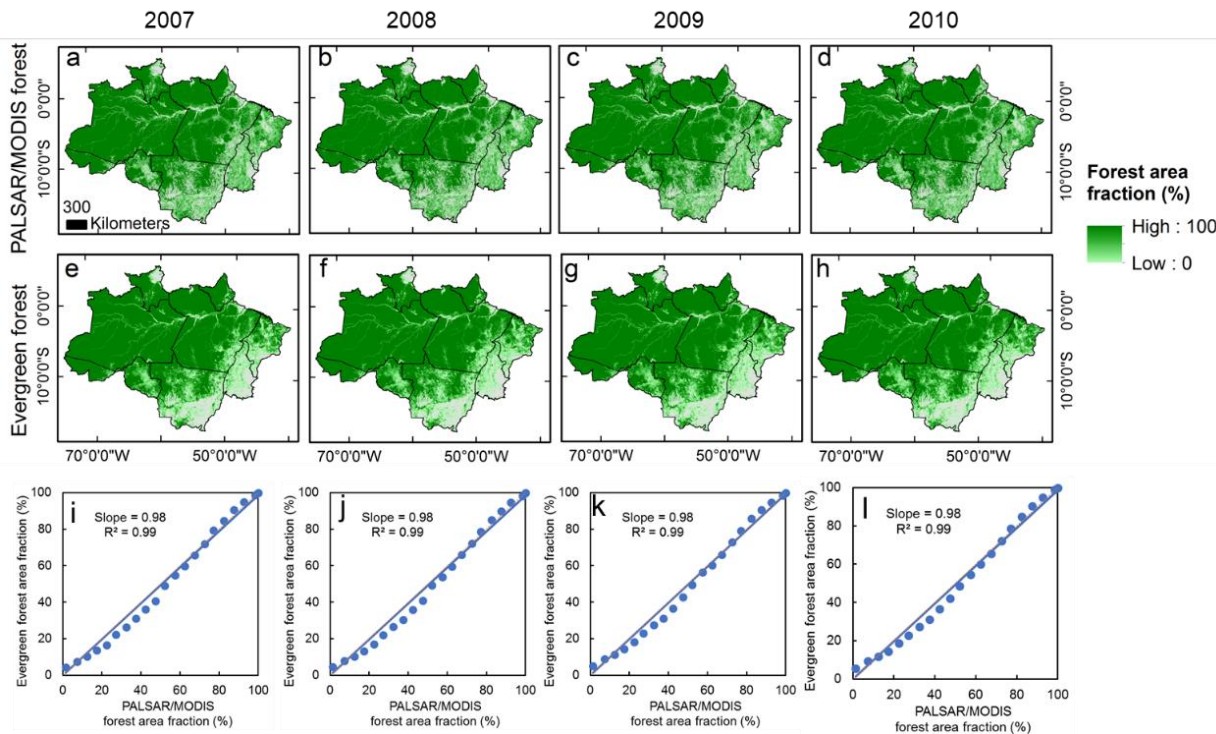

**Fig. 7.** Spatial consistency between annual PALSAR/MODIS forest cover maps and annual MODIS evergreen forest cover maps during 2007-2010 at a spatial resolution of 5 km. (a-d) Annual PALSAR/MODIS forest cover maps. (e-h) Annual MODIS evergreen forest cover maps. (i-l) Linear regression analyses between the area fraction of MODIS evergreen forest and area fraction of PALSAR/MODIS forest for individual years (2007-2010) at a spatial resolution of 5 km (0.23 million pixels each year at a spatial resolution of 5 km). The average area fraction of forest was calculated at an interval of 5%.

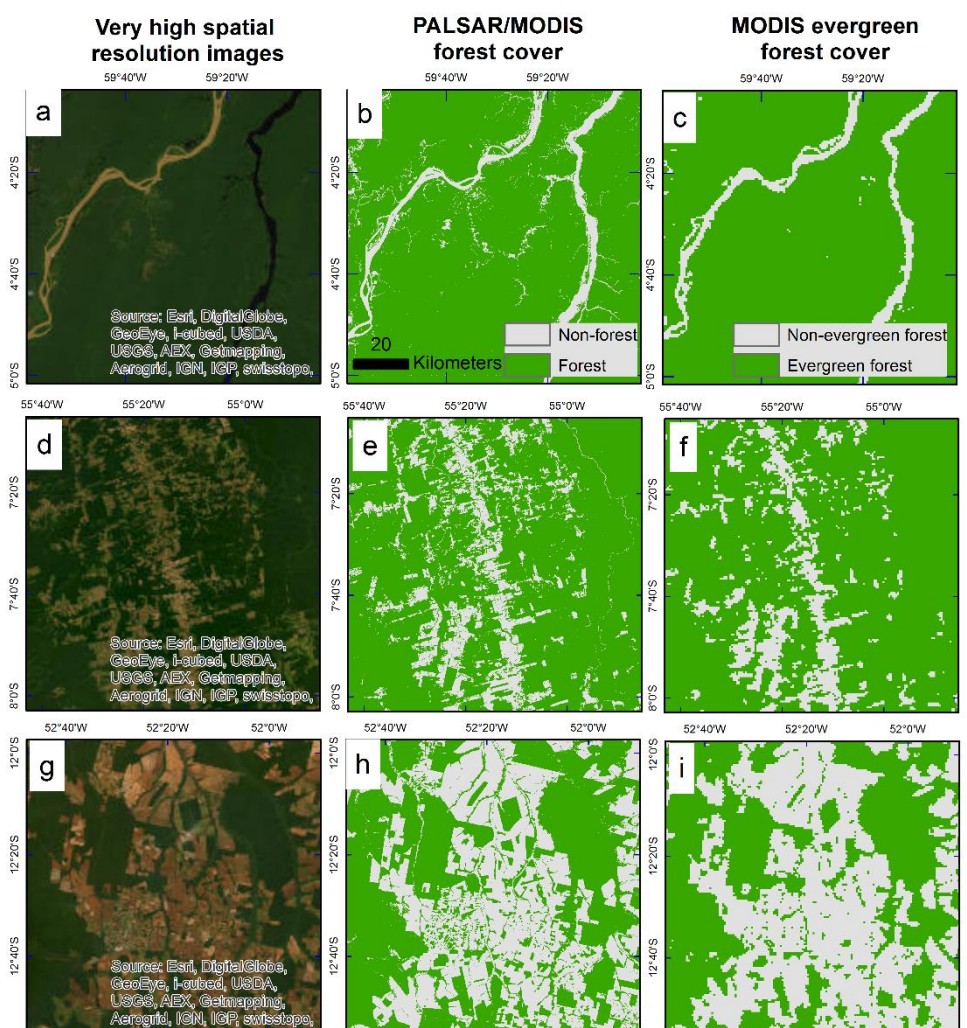

**Fig. 8.** Zoom-in windows of very high spatial resolution images, PALSAR/MODIS forest cover, and MODIS evergreen forest cover maps at three sites in the Brazilian Amazon. (a, d, and g) Very high spatial resolution images from ArcMap. (b, e, and h) PALSAR/MODIS forest cover maps. (c, f, and i) MODIS evergreen forest cover maps.

### 3.5. Annual forest area comparison in the Brazilian Amazon

A number of studies compared several forest cover maps derived from optical images and/or microwave images and reported that these forest cover maps have significant differences in forest area, spatial distributions, and temporal changes, which were attributed to the differences in forest definitions, satellite data, and forest mapping algorithms (Sexton et al., 2015; Qin et al., 2017). For example, one study compared eight previously satellite-based forest cover maps generated by optical images and found that global forest area ranged from $32.1 \times 10^6$ km$^2$ to $41.4 \times 10^6$ km$^2$, and claimed that one of the major reasons underlying the large discrepancy was the ambiguity in term of "forest" (Sexton et al., 2015). Frequent clouds and cloud

shadows substantially reduced the number of good-quality observations in optical images used to generate annual forest cover maps in the Brazilian Amazon (Qin et al., 2019). Our previous studies in Asia (Qin et al., 2015; Qin et al., 2016b) and South America (Qin et al., 2017) compared annual forest maps derived from (1) optical images only, (2) microwave images only, and (3) optical + microwave images, and concluded that the use of both optical and microwave images would substantially improve the accuracy of forest maps.

Our results here also show that forest area estimated in the Brazilian Amazon from the annual PALSAR/MODIS forest cover maps is consistent with that from the annual MODIS evergreen forest cover maps and the annual PALSAR-based forest cover maps developed by JAXA (Shimada et al., 2014) with both RMSE values of $0.03 \times 10^6$ km$^2$ during 2007-2010 (Fig. 9). Forest area estimated from annual PALSAR/MODIS forest cover maps and annual MODIS evergreen forest cover maps are slightly smaller ($0.3 \times 10^6$ km$^2$) than that from the Landsat-based Global Forest Watch (GFW) dataset (Hansen et al., 2013) generated using multiple-year Landsat 7 images around 2010. The forest area estimated from the PRODES project (INPE, 2023) is much smaller than the area estimated by GFW, PALSAR/MODIS, and MODIS evergreen forest cover maps, as the PRODES project is focused on primary forests and deforestation in primary forest areas. Only one or two Landsat-like images that were as cloud-free as possible in the dry season, and were used to generate annual PRODES forest cover maps (INPE, 2023). Because of large areas of cloud coverage, annual estimates of forest areas from the PRODES showed large interannual variation (Qin et al., 2019). Our work again demonstrated the importance and potential of L-band microwave remote sensing and daily optical remote sensing images in forest cover mapping.

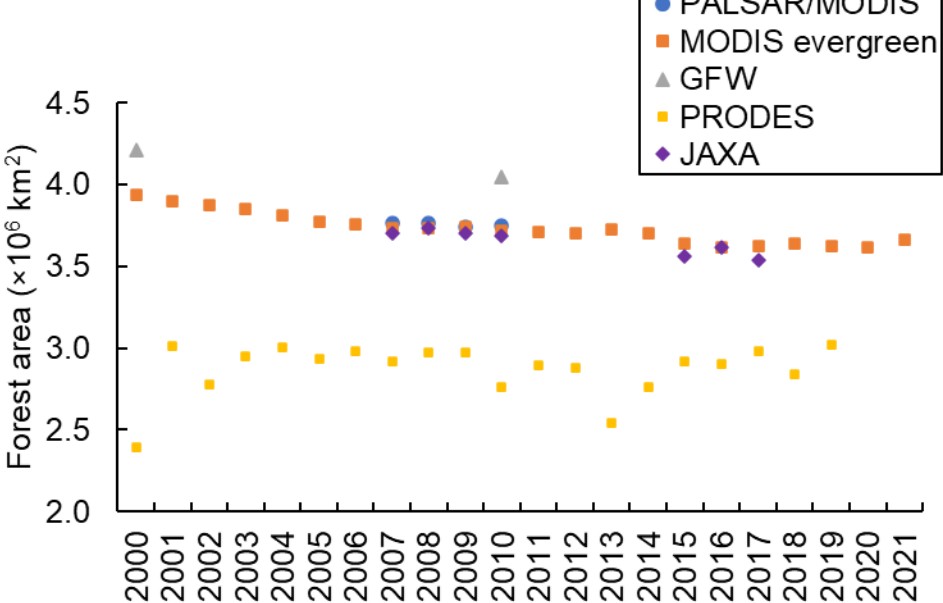

**Fig. 9.** Annual forest area estimated in the Brazilian Amazon during 2000-2021 from five forest cover data products. The PALSAR/MODIS forest areas and MODIS evergreen forest areas were calculated by this study. GFW forest areas were

calculated from the annual Global Forest Watch forest cover data in 2000 and 2010. PRODES forest areas were calculated from the annual PRODES forest cover maps. JAXA forest areas were calculated from the annual JAXA PALSAR-based forest cover maps.

390

## 4. Code and Availability

The annual forest and evergreen forest maps codes are available at the figshare (https://doi.org/10.6084/m9.figshare.21445626) (Qin and Xiao, 2022a). The annual forest maps (2007-2010) and evergreen forest maps (2000-2021) in the Brazilian Amazon have been submitted to the figshare data repository (https://doi.org/10.6084/m9.figshare.21445590) (Qin and Xiao, 2022b) in a GeoTIFF format. The data are provided in the spatial reference of South_America_Albers_Equal_Area_Conic.

## 5. Conclusion

Accurate forest cover maps are critical for tracking rapid forest changes and forest resource management. We generated annual PALSAR/MODIS forest cover maps and annual evergreen forest cover maps in the Brazilian Amazon from 2000 to 2021 using the FAO's forest definition as the reference. We then assessed the accuracy of the PALSAR/MODIS forest cover maps and evergreen forest cover maps using 1.1 million footprints of canopy height and canopy coverage datasets from ICESat/GLAS. We also compared the reasonability of the evergreen forest maps using the PALSAR/MODIS forest cover maps, which are little affected by the frequent clouds. The accurate PALSAR/MODIS forest cover maps and the evergreen forest cover maps could be used to understand better the interactions between forest and human activities and natural disturbances (Qin et al., 2019; Qin et al., 2021), which is vital to the forest resource management and forest conservation in the Brazilian Amazon. In the future, two recently launched platforms, the ICESat-2 satellite launched on September 15, 2018 (Markus et al., 2017) and the Global Ecosystem Dynamics Investigation (GEDI) instrument housed on the International Space Station (Dubayah et al., 2020), provides data that can substantially improve forest cover maps by providing Lidar-based canopy height and canopy coverage data to the public.

**Acknowledgments:** This study was supported in part by research grants from NASA Land Use and Land Cover Change program (NNX14AD78G), NASA Geostationary Carbon Cycle Observatory (GeoCarb) Mission (GeoCarb Contract # 80LARC17C0001), and NSF EPSCoR project (IIA-1946093, IIA-1920946).

**Author contributions:** X.X. and Y.Q. designed the study. Y.Q. and X.X. analysed the data, interpreted the results, and drafted the manuscript. H.T. and R.D. provided the ICESat canopy height and coverage data. All authors reviewed and edited the manuscript.

**Competing interests:** The authors declare no competing interests.

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
