# Peer review of "Yuanwei Qin1, Xiangming Xiao1\*, Hao Tang2, Ralph Dubayah3, Russell Doughty4, Diyou Liu5, Fang Liu1, Yosio Shimabukuro6, Egidio Arai6, Xinxin Wang7, Berrien Moore III4"

_Earth System Science Data, 2022_

## Author Comment (AC4)

[Figure]

**Fig. 11.1** Map of Brazil showing the distribution of the deciduous forests (shaded areas) and the location of the Paranã River Valley in Central Brazil

Fig. 11.1 in the book edited by Scariot et al.

---

## Author Response (AR1)

**Response to the reviewers' comments**

We appreciate the time and efforts of the three reviewers to provide these valuable comments and suggestions to improve this manuscript. To help reviewers see our responses to the comments and suggestions, we labeled all the comments and suggestions at the corresponding locations and used the Track Change mode to show all the revisions in the revised manuscript. According to these comments and suggestions, we revised the manuscript carefully. Please see our point-to-point responses to each comment and suggestion.

RC1: 'Comment on essd-2022-379', Anonymous Referee #1, 12 Apr 2023

As per the abstract, the article discusses the discrepancies in forest cover maps generated using different forest definitions and validation data which is affected by two biophysical parameters, tree height and canopy coverage. However, the paper presents the annual and evergreen forest cover maps in the Brazilian Amazon from 2000 to 2021 using PALSAR/MODIS and MODIS. The maps are evaluated using ICESat/GLAS as a reference for canopy height (>5m) and canopy cover (>10%) which is based on the general forest definition.

In general, it is challenging to map forests or trees that are less than 5m using satellite data, whether optical or SAR. Therefore, obtaining 98.5% of forest pixels with canopy height greater than 5 meters in both PALSAR/MODIS and MODIS is not unexpected. Consequently, the author should reconsider the main question of the paper, which should not be solely focused on forest definition but should instead aim to validate the annual forest cover maps. The abstract of the paper needs to be revised since the paper does not emphasize the significant discrepancies among the forest cover maps according to the FAO definition.

Response: Thanks for your time and efforts to help improve this manuscript. We revised the manuscript and focused on the annual PALSAR/MODIS forest cover maps, annual MODIS evergreen forest cover maps, and the assessment of these annual forest cover maps and annual evergreen forest cover maps in the Brazilian Amazon. We also revised the abstract according to your comments.

"**Abstract.** Many forest cover maps have been generated by using optical and/or microwave images, but these forest cover maps have large area and spatial discrepancies. To date, few studies have assessed forest cover maps in terms of two biophysical parameters used in forest definition: (1) canopy height and (2) canopy coverage. We generated annual forest cover maps from 2007 to 2010 and evergreen forest cover maps from 2000 to 2021 in the Brazilian Amazon using the images from the Phased Array type L-band Synthetic Aperture Radar and the time series images from the Moderate Resolution Imaging Spectroradiometer, using the forest definition of the Food and Agriculture Organization (FAO) of the United Nations (> 5-m tree height and > 10% canopy coverage) as the reference. We used the canopy height and canopy coverage datasets from the Geoscience Laser Altimeter System during 2003-2007 to assess annual forest cover maps from 2007 to 2010 and annual evergreen forest cover maps from 2003 to 2007, and the results show high accuracy of these forest cover and evergreen forest cover maps in the Brazilian Amazon. These annual forest cover maps and annual evergreen forest cover maps provide data support for the analyses of the causes, processes, and consequences of forest cover changes in the Brazilian Amazon."

Specific comments:

1. Some of the figures presented in the paper do not provide substantial information and could be included as supplementary material instead. For instance, Figure 4a and 4b lack distinct spatial information, which could be conveyed by a histogram (4c and 4d) or just by showing the range of values.

Response: We moved Figure 3 and Figure 4 into supplementary material.

2. The author stated in line 91 and in Fig 9 that the PALSAR/MODIS forest cover maps and MODIS evergreen forest cover maps had a close to 1:1 linear relationship at a 5-km spatial resolution. Is this because the PALSAR forest cover maps also incorporated MODIS NDVI? If so, does integrating PALSAR/MODIS offer any significant benefits over only using MODIS except for availability of cloud-free images?

Response: Brazilian Amazon is dominated by evergreen forest. This is the major reason that PALSAR/MODIS forest cover maps had a close to 1:1 linear relationship with MODIS evergreen forest cover maps at a 5-km spatial resolution.

As mentioned by the reviewers, the frequent clouds and cloud shadows reduced the availability of cloud-free optical satellite images, making forest cover mapping a challenging task in the Brazilian Amazon. Compared to optical satellite images, the L-band microwave PALSAR images had two following advantages in forest cover mapping: (1) the PALSAR images are independent of clouds and cloud shadows and (2) the electromagnetic wave of PALSAR has strong penetration capability into forest canopy and interacts with tree branches and trunks, thus, PALSAR data can be good indicators of forest structure and aboveground biomass and has strong backscatter signals, while optical satellite images have little penetration capability and mainly observe the top canopies of trees. The use of MODIS NDVI in the PALSAR/MODIS forest cover mapping was to reduce the commission error from urban, bare soil and rocks, which have similar PALSAR backscatter features but low to no vegetation coverage.

In this study, we used the PALSAR/MODIS forest cover maps as the reference maps, and then we compared and confirmed the reliability of the annual MODIS evergreen forest cover maps in the Brazilian Amazon. Both PALSAR/MODIS forest cover maps and MODIS evergreen forest cover maps matched very well, showing that the combination of PALSAR and MODIS images, and dense time series MODIS images can both generate high-accuracy forest cover maps in the Brazilian Amazon.

We added more details in the Introduction section (Page 2) and Section 3.3 (Page 14).

Minor

1. Citation of fao should be FAO, fra should be FRA.

Response: Revised as you suggested.

**RC2**: ['Comment on essd-2022-379'](), Anonymous Referee #2, 11 Jun 2023

General comments:

The manuscript is generally well written, the topic is important, and the outputs are useful for readers. However, I found two serious flaws (see below), both of which are related to the main contribution of this study (i.e., forest cover mapping following the FAO's definition of "forest"). Because they are fatal as a scientifically robust dataset, I must recommend this manuscript against publication.

1. This study uses the FAO FRA definition of forest only in part: "canopy height of > 5 meters and canopy coverage of > 10%" (P5L169-170). It ignores the other part of the definition "land with a minimum area of 0.5 hectares (with trees)" (P5L170) in the data analysis. The annual PALSAR/MODIS forest cover maps were produced at 50-m spatial resolution (L189-190) that is equivalent to 0.25 ha per pixel. Two or more adjacent pixels are necessary to be classified as "forest" under the FAO FRA definition. The data analysis does not take this into account.

Response: We appreciate the reviewer's comment. A minimum area of 0.5 ha in the FAO forest definition can be considered as the mapping unit. One can convert the raster data (50-m pixel) into a polygon map, and then calculate the polygon sizes and select those polygons that are 0.5 ha or larger for forest. People can use our data product at 50-m spatial resolution for such tasks.

FAO defines forest as land with a minimum area of 0.5 ha with (1) a tree canopy height of > 5 m and (2) a canopy coverage of >10% at the time of observations (land cover perspective), and it also includes lands with trees that can reach these thresholds at the time of tree mature (land use perspective). Forest remote sensing mainly focuses on the forest cover. The minimum land area of 0.5 ha is a precondition for forest but is impacted by various tree distribution patterns or complex terrains. Besides, satellite images do not have the exact size of 0.5 ha. We revised Section 2.5 as

"2.5. FAO forest definition

Hundreds of different forest definitions have been used in forest management (Unfao, 2002; Lund, 2014). FAO defines forest as land with a minimum area of 0.5 hectares with (1) a tree canopy height of > 5 m and (2) a canopy coverage of >10% at the time of observations, and it also includes lands with trees that can reach these thresholds at the time of tree mature (Fra, 2020). Using the FAO's forest definition as the reference (a tree canopy height of > 5 m and a canopy coverage of >10%), we identified and generated annual PALSAR/MODIS forest cover maps in the Brazilian Amazon during 2007-201. Then, we defined evergreen forests as forests with green leaves year-round, a tree canopy cover more than 10%, and a tree height larger than 5 m. and generated annual MODIS evergreen forest cover maps from 2000 to 2021. Note that as we use satellite images to identify and map forests, we do not consider lands with trees that can reach these two thresholds at the time of tree mature. Due to various spatial resolutions of satellite images, tree distribution patterns, and terrains, the minimum forest mapping area may not be exactly 0.5 hectares."

2. "Forest" and "evergreen forest" are not identical in definition. According to Scariot et al. (2008), deciduous forests are considerably distributed in the eastern part of the study area. When the authors want to produce the "forest" and "evergreen forest" cover map, they must exclusively use the PALSAR/MODIS data products acquired on the "leaf-on" and "leaf-off" seasons, respectively. The data analysis does not take this into account. They should also be careful to use the ICESat/GLAS data since the date of data acquisition can affect the accuracy of canopy height and coverage for deciduous tree species. I suspect the accuracy of forest cover maps produced through this study, because the estimated areas of PALSAR/MODIS "forest" cover ($3.77 \times 106$ km$^2$ in 2007 and $3.75 \times 106$ km$^2$ in 2010; P9L232) and MODIS "evergreen forest" cover ($3.73 \times 106$ km$^2$ in 2007 to $3.72 \times 106$ km$^2$ in 2010; P11L261) are nearly identical, even though different forest types are calculated.

Scariot, A., Vieira, D.L., Sampaio, A.B., Guarino, E., Sevilha, A. (2008). Recruitment of Dry Forest Tree Species in Central Brazil Pastures. In: Post-Agricultural Succession in the Neotropics. Springer, New York, NY. https://doi.org/10.1007/978-0-387-33642-8_11

Response: Ecologically, plants in high mountains and mid-to-high latitudes shed leaves in the winter because of cold air temperature (lower than zero degree centigrade). Tropical zones have few locations with air temperature of lower than zero degree centigrade. Thus, the area of deciduous trees in moist tropical zones should be small. Trees could shed leaves in dryland, too, because of extreme water stress. The Amazon basin generally has large annual rainfall.

PALSAR images have two following advantages in forest cover mapping: (1) the PALSAR images are independent of clouds and cloud shadows and (2) the electromagnetic wave of PALSAR has strong penetration capability into forest canopy and interacts with tree branches and trunks, thus, PALSAR data can be good indicators of forest structure and aboveground biomass and has strong backscatter signals in the "leaf-on" and "leaf-off" seasons. The use of the maximum MODIS NDVI each year in the PALSAR/MODIS forest cover mapping was to reduce the commission error from urban, bare soil and rocks, which have similar PALSAR backscatter features but low vegetation coverage.

We appreciate the reviewer sharing the book edited by Scariot et al. We read the book and found the figure for the spatial distribution (Fig. 11.1 in the book) of deciduous forest in Brazil. We did not find what the data source and approach were used to generate this deciduous forest map in Brazil. This deciduous forest map seems to be a sketch map of deciduous forest distribution. For the evergreen forest cover mapping in this study, we used all the 8-day MOD09A1 time series observations without clouds, cloud shadows, and snow in each year, including both "leaf-on" and "leaf-off" seasons as the reviewer recommended. Please see the details in section 2.7.

The ICESat had a revisit cycle of every 91 days. At each footprint, the maximum canopy height and canopy cover percentage were calculated from the ICESat LiDAR waveform signals acquired from 2003 to 2007 and screened for several confounding factors (e.g., cloud, noise, and topographic slope), which could reduce the impact of the date of data acquisition on the assessment of canopy height and coverage. Please see the details in "Section 2.4. ICESat canopy height and cover percentage data".

[Figure]

**Fig. 11.1** Map of Brazil showing the distribution of the deciduous forests (shaded areas) and the location of the Paranã River Valley in Central Brazil

Specific comments:

P1L1: Consider revising the title, since readers may misunderstand that this paper produced "annual forest" and "evergreen forest" cover maps.

Response: We revised the title to "Annual maps of forest and evergreen forest in the Brazilian Amazon from analyses of PALSAR and MODIS images".

P1L22-23, l26-27: Clearly distinguish the terms "annual forest cover" and "evergreen forest cover" in the text.

Response: We added the definition of forest and evergreen forest (Page 7) in the Methods section.

P1L32, P2L2, L4, L45: "Fao" > "FAO"

Response: Revised as you suggested.

P2L50: "Inpe, 2020" > Missing in the list of references.

Response: We added this reference.

P4L122: "MOD09A1 data product" > Which version of MOD09A1 was used?

Response: We added the version of the MOD09A1 (Collection 6).

P4L124: "Near-Infrared (NIR, 841 - 876 nm), NIR (1230 - 1250 nm)" > "Near-Infrared (NIR, 841 - 876 nm, and 1230 - 1250 nm)"

Response: Revised as you suggested.

P5L134: "blue, red, near infrared (841 – 875 nm), and shortwave infrared (1628 – 1652 nm) bands" > "Blue, Red, Near-Infrared (841 – 876 nm), and Shortwave Infrared (1628 – 1652 nm) bands" //Refer to P4L123-124.

Response: Revised as you suggested.

P5L138-139 (also P9L209-210, L214): "bad-quality", "good-quality" > "good" or "bad" sounds a subjective judgement by the author. Consider using the other word. Show the criteria of quality evaluation and explain what "the pixel quality layers" are in the manuscript.

Response: We revised the sentence on Page 5 as "We used the observations with the property of "VI produced with good quality" based on the MOD13Q1 quality band (DetailedQA) in time series analysis". We also revised the sentence on Pages 8-9 as "We calculated (1) the frequency (percentage) of the number of observations with LSWI $\geq 0$ over all the available observations ($FQ_{LSWI \geq 0}$) and (2) the minimum EVI values ($EVI_{min}$) in a year after excluding observations of clouds, cloud shadows, and snow based on the MOD09A1 quality band (StateQA)."

P5L153: "1.1 million footprints" > "$1.1 \times 10^6$ footprints"

Response: Revised as you suggested.

P5L158: "identified as forest footprints" > Please explain clearly how to identify tree canopy only from the LiDAR-based height data (and without spectral data on land cover).

Response: Both canopy coverage and canopy height were calculated from ICESat lidar waveform signals. We modified this sentence as "When considering both canopy height and canopy coverage, $0.9 \times 10^6$ footprints (85.1%) had a canopy height of $> 5$ m and canopy coverage of $> 10\%$, and these footprints were thus identified as forest footprints in terms of the FAO FRA forest definition."

P6L165 (Fig. 1): Show the spatial scale in each map. When is the date of data acquisition for each map?

Response: We added the spatial scale in the map and added the date of data acquisition in the caption.

P6L170: "Unfao" > "UNFAO" //Is this different from "FAO"?

Response: Revised as you suggested.

P6L172: "Fra" > "FRA"

Response: Revised as you suggested.

P6L173-174: "annual PALSAR/MODIS forest maps" > "annual PALSAR/MODIS forest cover maps", "annual evergreen forest maps" > "annual MODIS evergreen forest cover maps"

Response: Revised as you suggested.

P7L184: Describe which months "the dry season" are.

Response: Revised as you suggested.

P7L193-P8L200 (Figs. 2-4): Show the spatial scale in each map.

Response: Revised as you suggested.

PL206: Please specify the acquisition date of "MOD09A1 product" for each year and provide the spatial distribution visually as Fig.2.

Response: We used all the 8-day MOD09A1 observations in each year after excluding the observations covered by clouds, cloud shadows, and snow. Thus, we can have up to 46 observations in one pixel in a year. Thus, generating the spatial distribution of the acquisition date for MOD09A1 is challenging. In this revision, we revised the text and made it clear that we use all available 8-day MOD09A1 data in each year.

P9L207-208: "using the FAO's forest definition" > FRA (2020) does not define "evergreen forest". Clearly describe what the definition of "evergreen forest" is.

Response: We added the definition of evergreen forest as "We defined evergreen forests as the forests with green leaves year-round, a tree canopy cover more than 10%, and a tree height larger than 5 m." (section 2.5)

P9L216, P15L341: "forest mapping" > "forest cover mapping"

Response: Revised as you suggested.

P9L220: "during 2003-2005" > Are there any reasons why the author selected these years for the spatial comparison? If not, I suggest the author using the same period ("during 2007-2010") for both forest cover maps to make them comparable.

Response: The ICESat GLAS datasets were acquired during 2003-2007. To reduce the potential bias due to date acquisition, we chose the annual evergreen forest cover maps during 2003-2007 for the spatial comparison. We revised the manuscript and corrected the typo as "during 2003-2007".

P9L234, P11L265: "18 5" > "185" //Delete the space between "8" and "5".

Response: To avoid confusion, we rewrote the two sentences as "To assess the accuracy of the PALSAR/MODIS forest cover map, we used two independent reference datasets. First, we used the land cover maps at the 2-m spatial resolution from the Global Land Cover Validation Reference Dataset in 2010, which had land cover maps in 18 blocks ($0.15 \times 10^6$ pixels) in the Brazilian Amazon and each block covered an area of $5 \times 5$ km$^2$ (Olofsson et al., 2012; Stehman et al., 2012). Second, we used the land cover maps from the TREES-3 (Achard et al., 2014) reference dataset at the 30-m spatial resolution from the European Commission's Joint Research Centre (JRC), which had land cover maps in 416 blocks ($17.09 \times 10^6$ pixels) and each block covered an area of $10 \times 10$ km$^2$." (Page 9)

On page 12, we revised the sentence as "We previously assessed the accuracy of the evergreen forest cover map in 2010 using 18 blocks with $5 \times 5$ km$^2$ ($1.27 \times 10^3$ pixels) of land cover maps at 2-m spatial resolution from the Global Land Cover Validation Reference Dataset in 2010 and 416 blocks with $10 \times 10$ km$^2$ ($0.13 \times 10^6$ pixels) of land cover maps from the 30-m TREES-3 (Achard et al., 2014) reference dataset."

P11L253, P13L280 (Figs. 6, 8): Please explain from "a" to "l" shown in each map.

Response: Revised as you suggested.

P12L271 (Fig. 7): "a-v" > It is better to show the year on each map.

Response: Revised as you suggested.

P13L289-290: "There might be some non-evergreen forest and sparse forest in the Cerrado, which are not identified as forest in the MODIS evergreen forest cover maps." > This sounds an invalid interpretation by the authors because they did not consider the date of data acquisition (leaf-on/ leaf off seasons) when they selected the MOD09A1 data product.

Response: We used all 8-day MOD09A1 images after excluding clouds, cloud shadows, and snow to generate annual evergreen forest cover maps each year, including MOD09A1 images in leaf-on and leaf-off seasons.

P14L295-299 (Fig. 9): Low (0%) of the forest area fraction should show as white in color. Confirm if "0.23 million pixels" is correct, because PALSAR/MODIS and MODIS evergreen have different spatial resolutions.

Response: We showed the low- or non-forest area fraction as grey in color. We rewrote this sentence as "0.23 million pixels each year at a spatial resolution of 5 km".

P14L301-317: Section 3.4 is rather general in description and is neither the fact derived from the data analysis (result) nor an interpretation of the derived results (discussion). Move it to 1. Introduction or 2. Method. The latter part could be moved to 5. Conclusion (as future research direction). The author does not have to calculate the average area fraction of forest "at an interval of 5%".

Response: Revised as you suggested. We moved the materials in this section to the Introduction section, Methods section, and Conclusion section.

P16L342 (Fig. 10): "This study" > "MODIS evergreen"? Both "PALSAR/MODIS" and "This study" are the results derived from this study. Please explain each abbreviation. I found that the forest area increased from 2012 to 2013 according to "this study" (MODIS evergreen). Please explain where and why it occurred in the Brazilian Amazon.

Response: Revised as you suggested. We changed "This study" to "MODIS evergreen". We also explained each abbreviation in the Figure 10 title.

The MODIS evergreen forest cover maps in this study include primary forest and secondary forest. We calculated the evergreen forest changes between 2012 and 2013. We found that small forest area gains in the sparse forest and agriculture areas in the south and east of the Brazilian Amazon (Figure R2_1), which may be contributed by the secondary forest growth or tree plantations under favorable climate conditions (Figure R2_2) in 2013.

[Figure]

Figure R2_1. MODIS evergreen forest cover gain in 2013.

[Figure]

Figure R2_2. Difference of annual total precipitation between 2012 and 2013. The precipitation data is from the Tropical Rainfall Measuring Mission data product.

P17L372: Place this reference (FAO, 2020) in alphabetical order.

Response: Revised as you suggested.

P18L420: "PRODES" > Confirm if this reference is correctly listed.

Response: Revised as you suggested. We revised this reference as "INPE. PRODES-Amazon: http://www.obt.inpe.br/OBT/assuntos/programas/amazonia/prodes"

RC3: 'Comment on essd-2022-379', Anonymous Referee #3, 02 Jul 2023

This manuscript attempts to generate annual forest cover maps and annual evergreen forest cover maps using FAO's forest definition and combined application of Synthetic Aperture Radar (SAR) and optical (MODIS) images from 2000 to 2021. The working is meaningful and challenging, and the outputs may be useful for readers. Moreover, this manuscript is well-written in general. However, some details are neglected by the authors. Firstly, in the definition of forest in FAO, the "land with a minimum area of 0.5 hectares" is a precondition, but we can not see how the authors to deal with this precondition on the SAR and optical images. Secondly, some reasons and criterions are not given out in this manuscript although the authors may consider they are general conclusions in previous studies. For example, why do the authors select 5-km spatial resolution as an optimal scale to aggregate the 50-m SAR and 500-m MODIS images? Thirdly, there are little usefully statistical information in this manuscript, although a little bit of

summation of pixels for two types of forest dataset are addressed. This deficiency will raise the risk of incredibility of results. Finally, as commented by the referee 2, the definition and difference of "forest" and "evergreen" are not clearly addressed in this manuscript by authors, which might result in the area discrepancy between PALSAR/MODIS forest cover maps and MODIS evergreen forest cover maps. Specific comments are as follows.

Response: Thank you so much for your positive comments. We really appreciate your detailed comments to improve this manuscript and the manuscript writings.

We appreciate the reviewer's comment. A minimum area of 0.5 ha in the FAO forest definition can be considered as the mapping unit. One can convert the raster data (50-m pixel) into a polygon map, and then calculate the polygon sizes and select those polygons that are 0.5 ha or larger for forest. People can use our data product at 50-m spatial resolution for such tasks.

FAO defines forest as land with a minimum area of 0.5 hectares with (1) a tree canopy height of > 5 m and (2) a canopy coverage of >10% at the time of observations (land cover perspective), and it also includes lands with trees that can reach these thresholds at the time of tree mature (land use perspective). Forest remote sensing mainly focuses on the forest cover. The minimum land area of 0.5 hectares is a precondition for forest but is impacted by various tree distribution patterns or complex terrains. Besides, satellite images do not have the exact size of 0.5 hectares. We revised Section 2.5 as

"2.5. FAO forest definition

Hundreds of different forest definitions have been used in forest management (FAO, 2002; Lund, 2014). FAO defines forest as land with a minimum area of 0.5 hectares with (1) a tree canopy height of > 5 m and (2) a canopy coverage of >10% at the time of observations, and it also includes lands with trees that can reach these thresholds at the time of tree mature (FRA, 2020). Using the FAO's forest definition as the reference (a tree canopy height of > 5 m and a canopy coverage of >10%), we identified and generated annual PALSAR/MODIS forest cover maps in the Brazilian Amazon during 2007-201. We defined evergreen forests as forests with green leaves year-round, a tree canopy cover more than 10%, and a tree height larger than 5 m. Then we generated annual MODIS evergreen forest cover maps from 2000 to 2021. Note that as we use satellite images to identify and map forests, we do not consider lands with trees that can reach these two thresholds at the time of tree mature. Due to various spatial resolutions of satellite images, tree distribution patterns, and terrains, the minimum forest mapping area may not be exactly 0.5 hectares."

We added the criteria of MOD13Q1 (Page 5) and MOD09A1 (Page 5) data processing in the revised manuscript. We also added the reason that we selected 5-km spatial resolution as an optimal scale to aggregate the 50-m SAR and 500-m MODIS images (Page 9) "For the spatial comparison, to avoid the bias caused by different spatial resolutions, we aggregated the 50-m annual PALSAR/MODIS forest cover maps and 500-m (463-m) MODIS evergreen forest cover maps into 5-km pixels and calculated their average forest area fraction values within individual 5-km pixels.".

We revised section "2.8. Spatial and statistical analysis" and added the statistical analyses used in this study, including (1) the evaluation of the annual PALSAR/MODIS forest cover maps and annual MODIS evergreen forest cover maps based on the ICESat data, (2) the spatial comparison between the PALSAR/MODIS forest cover maps and MODIS evergreen forest

cover maps as well as their linear relationship and significant analyses, and (3) the forest area comparison based on the Root Mean Square Error (RMSE) between the PALSAR/MODIS forest cover maps, MODIS evergreen forest cover maps, and other three forest cover datasets.

We added the definition of evergreen forest in the revised manuscript (Page 7) as "We defined evergreen forests as forests with green leaves year-round, a tree canopy cover more than 10%, and a tree height larger than 5 m."

Please see our responses to the specific comments as follows.

P1, line 25, insert a space between ">" and numbers.

Response: Revised as you suggested.

P1, line 31, suggest changing "$40.6 \times 10^6$" to "$4.1 \times 10^7$", which is in line with the scientific expressions in the next pages.

Response: We used the multiplication step for increments of the base-10 exponent of three to match thousand, million et al.

P1, line 32, change "Fao" to "FAO", and check it through this manuscript (P2, line 36; P2, line 38; P2, line 45) and revise it.

Response: Revised as you suggested.

P1, line 34, insert "of" before "aboveground".

Response: Revised as you suggested.

P2, line 35, insert "the" after "in".

Response: Revised as you suggested.

P2, line 38, insert "has" after "loss".

Response: Revised as you suggested.

P2, line 40, change "people" to "us".

Response: Revised as you suggested.

P2, lines 49, 54, insert "a" before "high".

Response: Revised as you suggested.

P2, line 51, change "free cloud" to "cloud-free".

Response: We revised this sentence as "Google Earth Engine, a powerful cloud computing platform, was also developed to process these big datasets".

P2, line 56, change "shadow" to "shadows".

Response: Revised as you suggested.

P2, line 62, change "non-forest" to "non-vegetation".

Response: Revised as you suggested.

P2, line 65, change "reduced" to "reducing", change the first "and" to "but".

Response: Revised as you suggested.

P3, lines 71-72, 77, change "very high" to "higher".

Response: Revised as you suggested.

P3, line 73, change "a fixed locations" to "in situ".

Response: Revised as you suggested.

P3, line 79, delete "Tree" and change "canopy" to "Canopy".

Response: Revised as you suggested.

P3, line 96, change "Amapa", "Para", "Maranhao" and "Rondonia" to "Amapá", "Pará", "Maranhão" and "Rondônia".

Response: Revised as you suggested.

P3, line 97, insert a space after "world".

Response: Revised as you suggested.

P4, line 103, insert a space after "decades".

Response: Revised as you suggested.

P4, line 108, please give the full name of JAXA (Japan Aerospace Exploration Agency) in the first time.

Response: Revised as you suggested.

P4, line 109, insert a space between left bracket and right bracket.

Response: Revised as you suggested.

P4, lines 111-112, suggest moving "further" before "provided".

Response: Revised as you suggested.

P4, line 122, please give the version of MOD09A1. V061?

Response: Revised as you suggested.

P4, line 124, same as shortwave infrared, merge two ranges of NIR in one bracket, "Near-Infrared (NIR, 841-876 nm and 1230-1250 nm)".

Response: Revised as you suggested.

P4, line 130, insert a space between left bracket and right bracket.

Response: Revised as you suggested.

P5, line 134, change "875" to "876".

Response: Revised as you suggested.

P5, lines 134-135, change the hyphen into English status of "-".

Response: Revised as you suggested.

P5, line 138, change "exclude" to "excluded", please use the past tense.

Response: Revised as you suggested.

P5, line 139, change "use" to "used".

Response: Revised as you suggested.

P5, lines 145, 147, 151, insert a space before left bracket.

Response: Revised as you suggested.

P5, line 153, change "million" to "$\times\ 10^6$".
Response: Revised as you suggested.

P5, line 156, delete "of the", and insert "a" before "canopy height".
Response: Revised as you suggested.

P5, lines 155, 157, 159, change "meters" to "m".
Response: Revised as you suggested.

P5, line 160, insert "canopy" before "coverage".
Response: Revised as you suggested.

P5, line 161, insert "in the" before "north".
Response: Revised as you suggested.

P5, line 162, insert a comma before "south".
Response: Revised as you suggested.

P6, line 170, insert a space after the second "of" and change "meters" to "m".
Response: Revised as you suggested.

P6, line 172, insert a space after "mature", and change "Fra" to "FRA".
Response: Revised as you suggested.

P6, line 178, insert a space after the first "data", and change "PALSAR data" to "Electromagnetic wave of PALSAR".
Response: Revised as you suggested.

P7, line 194, change "during 2007-2010. (a) 2007. (b) 2008. (c) 2009. (d) 2010." to "during 2007 (a), 2008 (b), 2009 (c), and 2010 (d)." in the caption of figure 2.
Response: Revised as you suggested.

P8, line 199, Figure 4a, b, change "gamma0" to "gamma-naught", which is in line with the term emerging at the page 7, line 183.

Response: Revised as you suggested.

P8, line 200, change "gamma0 (a-b) and their histograms (c-d)" to "gamma- naught (a, b) and their histograms (c, d)" in the caption of figure 4.

Response: Revised as you suggested.

P9, line 214, change "included" to "conducted".

Response: Revised as you suggested.

P9, line 217, change "image" to "images".

Response: Revised as you suggested.

P9, line 224, change "that have ICESat/GLAS footprint data" to "that contained part information of ICESat/GLAS footprint data".

Response: Revised as you suggested.

P9, why do the authors choose the spatial resolution of 5-km? what is the reason? Why not 1-km? please give the citation or criterion.

Response: The MODIS data used in this study has a spatial resolution of 463 m (roughly 500 m). To reduce the bias for forest area comparison caused by the different spatial resolutions, we chose the spatial resolution of 5-km.

P9, line 228, change "in" to "during".

Response: Revised as you suggested.

P9, lines 219-228, in this section, we cannot conclude more useful statistical information for example significance of difference, although a little bit of summation of pixels for two types of forest dataset.

Response: We revised this section and added the statistical analyses we did in this study (Page 9).

P9, line 231, change "in" to "during".

Response: Revised as you suggested.

P9, line 234, insert a space after "disturbances".

Response: Revised as you suggested.

P9, lines 234, 236, in order to avoid misreading by the reader, suggest place the phrases of "5 × 5 km$^2$" and "10 × 10 km$^2$" after "blocks" and insert "with" before these two phrases.

Response: Revised as you suggested.

P9-10, lines 235, 236, 238, insert a space before the left bracket of citations.

Response: Revised as you suggested.

P9-10, lines 234-238, this long sentence confuses me a lot. Firstly, I think this is an incomplete sentence. What do the authors want to do using blocks of two reference datasets? Secondly, why the sparse spatial resolution maps have much more blocks compared with the higher resolution maps with few blocks. Do the authors select different sizes of hotspot for these two datasets to compare them with the forest cover maps generated in this study? Finally, please revise this sentence and make it easier understand.

Response: We revised this sentence and made it easier understand "To assess the accuracy of the PALSAR/MODIS forest cover map, we used two independent reference datasets. First, we used the land cover maps at the 2-m spatial resolution from the Global Land Cover Validation Reference Dataset in 2010, which had land cover maps in 18 blocks (0.15 × 10$^6$ pixels) in the Brazilian Amazon and each block covered an area of 5 × 5 km$^2$ (Olofsson et al., 2012; Stehman et al., 2012). Second, we used the land cover maps from the TREES-3 (Achard et al., 2014) reference dataset at the 30-m spatial resolution from the European Commission's Joint Research Centre (JRC), which had land cover maps in 416 blocks (17.09 × 10$^6$ pixels) and each block covered an area of 10×10 km$^2$. The overall accuracy of the PALSAR/MODIS forest cover map in 2010 was about 91% (Qin et al., 2019)."

P10, lines 240-241, the same comment as P7, line 194.

Response: Revised as you suggested.

P10, lines 245-246, please add the number of pixels of canopy height > 5 m and canopy coverage > 10%. Otherwise, delete the number of pixels in the brackets at line 248.

Response: Revised as you suggested.

P10, lines 248-249, insert a space between "of" and ">".

Response: Revised as you suggested.

P10, line 249, change "in" to "during". Please be more careful about the conclusion of "consistent with the FAO forest definition". As described in section 2.5. FAO forest definition, "land with a minimum area of 0.5 hectares" is the precondition.

Response: Revised as you suggested. We also revised section 2.5. FAO forest definition and made a careful conclusion.

P10, line 250, insert "that" before "in".

Response: Revised as you suggested.

P11, line 260, change "in" to "during".

Response: Revised as you suggested.

P11, line 262, change "estimates" to "estimated".

Response: Revised as you suggested.

P11, line 265, express the numbers of pixels using scientific notation for example power of ten in the brackets.

Response: Revised as you suggested.

P12, line 266, the same comment as p11, line 265.

Response: Revised as you suggested.

P12, lines 267-268, insert a space before the left bracket of citations.

Response: Revised as you suggested.

P12, line 275, insert a space between "of" and ">", and change "in" to "during".

Response: Revised as you suggested.

P13, line 277, change "in" to "during", and insert a space between "of" and ">". Be careful about the conclusion of "consistent with….".

Response: Revised as you suggested.

P13, line 280, change "canopy height and coverage" to "canopy height (m) and canopy coverage (%)".

Response: Revised as you suggested.

P13, line 284, change "in" to "during".

Response: Revised as you suggested.

P13, line 287, insert "a" before "good", and change "in" to "for".

Response: Revised as you suggested.

P13, line 288, insert "that of" after "than".

Response: Revised as you suggested.

P13, line 289, change the second "in the" to comma.

Response: Revised as you suggested.

P13, line 290, insert "area" after "Cerrado".

Response: Revised as you suggested.

P14, line 292, insert "the" after "at".

Response: Revised as you suggested.

P14, line 302, insert a space between "features" and left bracket.

Response: Revised as you suggested.

P14, line 303, change "estimates" to "estimated".

Response: Revised as you suggested.

P14, line 304, insert "canopy" before "coverage" and "that of" after "than".

Response: Revised as you suggested.

P14, lines 304-305, change "showed" to "shows".

Response: Revised as you suggested.

P14, line 309, insert a space between "disturbances" and left bracket.

Response: Revised as you suggested.

P15, lines 315-316, insert a space before the left bracket of citations.

Response: Revised as you suggested.

P15, line 321, insert a space between "algorithms" and left bracket.

Response: Revised as you suggested.

P15, line 322, change "previous" to "previously".

Response: Revised as you suggested.

P15, line 324, change "in the term" to "in term of", and insert a space between "forest" and left bracket.

Response: Revised as you suggested.

P15, line 325, change "reduce" to "reduced".

Response: Revised as you suggested.

P15, lines 326-327, insert a space before left bracket of citations.

Response: Revised as you suggested.

P15, lines 330, 333, change "estimates" to "estimated".

Response: Revised as you suggested.

P15, line 331, change "are" and "those" to "is" and "that", respectively.

Response: Revised as you suggested.

P15, line 332, insert a space between "JAXA" and left bracket.

Response: Revised as you suggested.

P15, line 333, insert "forest cover" between "PALSAR/MODIS" and "maps".

Response: Revised as you suggested.

P15, line 334, insert a space between "dataset" and left bracket.

Response: Revised as you suggested.

P15, line 335, give the full name of "PRODES", and insert a space between "project" and left bracket.

Response: Revised as you suggested.

P15, lines 335-336, change "estimates" to "estimated".

Response: Revised as you suggested.

P15, line 338, insert ", and" after "dry season", and insert a space between "maps" and left bracket.

Response: Revised as you suggested.

P15, line 340, insert a space between "variation" and left bracket, and change "demonstrates" to "demonstrated".

Response: Revised as you suggested.

P16, line 342, suggest change "million ha" in the label of y-axis to scientific notation and convert the unit of "ha" to "$km^2$".

Response: Revised as you suggested.

P16, line 343, change "estimates" to "estimated" in the caption of figure 10.

Response: Revised as you suggested.

Figures: all the maps in this manuscript are lack of spatial scale.

Response: We provided the spatial scale for all the maps in the revised manuscript.

---

## Referee Report (RR1)

**General comments:**

This study presented a datasets of forest map according to FAO's definition, which is of great importance to the sustainable development and biomass estimation in specific. From the results presented in this paper, the accuracy is of good performance, and it has a great contribution for large-scale forest mapping work. However, I recommend the authors to address the following issues before the manuscript can be considered to be published on ESSD.

1. The description of method about forest mapping is not comprehensive enough, even was ignored. Please express this part fully. The detail of methodology is missing, which is not acceptable without further modification regarding the scope of ESSD journal. It is not easy for readers to know how they produced the datasets without further knowledge from the cited paper.

2. The availability of existing reference datasets needs to be further clarified, especially in the accuracy comparison part.

3. For the statistical accuracy analysis, the authors used 5km as the pixels' resolution. Why not conduct such analysis on a higher resolution? The resolution of the cartographic results in this paper is already very low (e.g., 50 m, 250 m, 500 m). Using a lower resolution to is actually much more friendly to a better accuracy, but it's fake. Please reconduct the uncertainty analysis to make it more convincible like previous studies [1-4].

4. Did the forest map from 2007 to 2010 (Fig. 5) also combine PALSAR data and MODIS? Please clarify it, because the "or" in the caption is not consistent with the description in the text. For 2007-2021 evergreen forest mapping, only MODIS data were used only, which is clear. Similar mistakes are quite often in the manuscript, such as Sections 3.1 and 3.2. Please recheck the manuscript thoroughly.

5. Please add zoomed images (e.g., optical images) for details analysis to prove your results is corrected. Please refer to the existing research analysis [1-4].

6. Section 3.4 should be placed in the introduction instead of "Results and discussion" since section 3 described the advantages satellite lidar data for result assessment.

7. The comprehensive discussion of the results is inadequate in the manuscript, such as the possible reasons for misclassifications.

8. Generally speaking, the source data used in this paper are quite outdated. (e.g.,? This deficiency leads to a much lower significance of this study. Why not use more modern data like Sentinel, ICESat-2, ALOS-2? This should be answered seriously, as well as in the text.

10. The topic of this study, which is "annual forest" and "evergreen forest", as we can see in the title and abstract. But as I know, and as we can know from the results, these two types of forests are almost the same in Amazon. So why separate them apart if the difference is not significant (see Section 3.3 and Fig. 9)? Or, what's the difference between them and how we can tell it from the results?

**Reference:**

[1] Shimada, Itoh, Motooka, et al. New Global Forest/Non-Forest Maps from ALOS PALSAR Data (2007-2010) [J]. Remote Sensing of Environment, 2014, 2014,155(-): 13-31.DOI:10.1016/j.rse. 2014.04.014.

[2] Martone M, Rizzoli P, Wecklich C, et al. The global forest/non-forest map from TanDEM-X interferometric SAR data[J]. Remote Sensing of Environment, 2018, 205:352-373.DOI:10.1016/j.rse.2017.12.002.

[3] Mazza A, Sica F, Rizzoli P, et al.TanDEM-X Forest Mapping using Convolutional Neural Networks[J].Remote Sensing, 2019, 11(2980).DOI:10.3390/rs11242980.

[4] Pulella A,  Santos R A, Sica F, et al. Multi-Temporal Sentinel-1 Backscatter and Coherence for Rainforest Mapping[J]. Remote Sensing, 2020, 12(5):847-.DOI:10.3390/rs12050847.

**Specific comments:**

1. Line 262: $0.03 \times 10^6$ -> $3 \times 10^4$;

Line 248: $0.75 \times 10^6$ -> $7.5 \times 10^5$; please check such representation.

2. Line 165, Fig.1(a) and (b): The legend should be included in the figure.
3. Line 199, Fig.4(c) and (d): The label is too crowded to read.

4. Line 269, Fig.7: add a legend, like Fig. 5.

---

## Author Response (AR2)

**Response to the Comments from Editors and Reviewers**

**Editor comments**

Dear Authors,

While you have addressed some reviewers' concerns, I find myself in agreement with the comments put forth by Reviewer #3 and Reviewer #4. Particularly, their significant concerns regarding the lack of acknowledgment for existing methodologies and the absence of uncertainty analysis stand out.

Response: Dear Dr. Sasaki, We appreciate your time and efforts to help improve this manuscript. According to the comments from you and four reviewers, we made careful improvements to this manuscript in this third round of revision. We (1) added the detailed method description for forest cover and evergreen forest cover mapping in the Methods section and (2) added the error matrices (Table S1) based on two independent reference datasets and the comparison of very high spatial resolution images, PALSAR/MODIS forest cover maps, and MODIS evergreen forest cover maps at three selected sites (Fig. 8) in the Brazilian Amazon.

Additionally, I would like to raise the following points:

1) The terminology "annual forest cover" and "evergreen forest cover maps" as used in your manuscript remains ambiguous. If your study region encompasses X types of forest cover, it's crucial that the term "forest" adequately reflects those X types. To illustrate, at L155, you mentioned, ""The two major biomes in Brazil are the Amazon evergreen forests in the north and west, and the Cerrado, i.e., a vast ecoregion of tropical savanna, in the south and east." Based on this description, your annual forest would comprise both Amazon evergreen forests and tropical savannas. However, your paper seems to predominantly focus on the evergreen forest. If this is the intent, I suggest amending your title, abstract, and pertinent sections accordingly.

Response: For the long term, forest cover maps in the Brazilian Amazon were produced mainly based on optical remote sensing images from multiple years to get enough cloud-free images. Microwave remote sensing images are independent of frequent clouds in the tropics. The combination of microwave and optical remote sensing showed advantages in mapping annual tropical forest cover. Thus, we produced the annual PALSAR/MODIS forest cover maps in the Brazilian Amazon. These PALSAR/MODIS forest cover maps could include evergreen, deciduous, and mixed forests in the two major biomes. Evergreen forest cover is the dominant forest cover type in the Brazilian Amazon. Thus, PALSAR/MODIS forest cover and MODIS evergreen forest cover maps are important datasets. Besides the independent reference datasets at the site scales, the PALSAR/MODIS forest cover maps provided regional references for the area and spatial comparisons of MODIS evergreen forest cover maps. Generating annual forest cover type maps in the Brazilian Amazon is on our research schedule, which will need extensive time

and effort for the field survey, collection and integration of high spatial resolution remote sensing images, algorithm calibration and validation, and accuracy assessment of forest cover type maps.

2) The definition of "forest" in your work doesn't distinguish between evergreen, deciduous, or mixed forests. Your rationale for embracing the FAO's definition for evergreen forests needs clearer elucidation.

Response: We investigated and quantified the threshold values for the evergreen forest mapping algorithm based on the training samples of evergreen forest in the Brazilian Amazon (Qin et al., 2019, Nature Sustainability). We modified the relevant description of the evergreen forest mapping in the section "2.7. Algorithm and data of annual MODIS evergreen forest cover maps during 2000-2021" as

"Based on the canopy phenology from analyses of time series water-related LSWI and greenness-related EVI calculated from all MOD09A1 data in each year, a novel, simple and robust algorithm was developed to generate annual maps of evergreen forests in the Brazilian Amazon using the FAO's forest definition as the reference and evergreen forest training samples (Xiao et al., 2009; Qin et al., 2019)."

[Figure]

Supplementary Fig. 14. Minimum Enhanced Vegetation Index (EVImin) and Percent observations with LSWI $\geqslant$ 0 out of all good observations and their standard deviations for evergreen forest (using the FAO's forest definition as the reference) and other land cover types (Others). This analysis is calculated based on the Ground sample blocks from the Global Land Cover Validation Reference Dataset and EVI and Land Surface Water Index (LSWI) from MOD09A1 dataset in 2010. (Qin et al., 2019, Nature Sustainability.)

I kindly request that you address the aforementioned concerns and consider resubmitting the paper.

Sincerely,
Nophea Sasaki

**Editor comments**

by Svenja Lange

Notification to the authors:
Please ensure that the colour schemes used in your maps and charts allow readers with colour vision deficiencies to correctly interpret your findings. Please check your figures using the Coblis – Color Blindness Simulator (https://www.color-blindness.com/coblis-color-blindness-simulator/) and revise the colour schemes accordingly.

Response: Dear Dr. Lange, Thanks for your suggestions. We checked all the figures and we did not use green/red colour schemes. We also made Fig. 4 and Fig. 6 bigger to help readers to see them clearly.

**Review #1**

accepted as is

Response: Thanks!

**Review #3**

The authors have carefully addressed my concerns in the revised manuscript. I would like to recommend this manuscript to be published in the ESSD after a few of technical corrections which also can be carried out at the phase of proofreading.

Response: Thanks!

**Reviewer #4**

This paper presents the generation of annual forest and evergreen forest cover maps in Brazilian Amazon by combining PALSAR and MODIS data. The manuscript is well written and easy to follow. However, I have several major concerns. Firstly, the same method and data have been used in several previous studies (Qin et al., 2016a; Qin et al., 2017; etc.). From this perspective, this study lacks novelty. Secondly, during the validation of forest maps, only the overall accuracy of 91% was mentioned. It would be better to include error matrices based on two reference data to provide a more complete and reliable assessment of mapping accuracy. Thirdly, as mentioned by the other reviewer, this paper lacks useful statistical analysis. It is advised that the authors quantify annual change (loss, gain, and net change) in forest cover in Brazilian Amazon and

identify the hotspots of forest cover change. Additionally, apart from comparing the annual forest area from different forest products in Figure 10, it would be beneficial to perform more localized visual comparisons of forest cover from current forest maps. Lastly, this project produced 50m forest maps from 2007 to 2010 using PALSAR and MODIS data. Since these two data sources have different spatial resolutions, how these multi-scale data were fused?

Response: Thanks for your time and efforts to improve this manuscript. We previously made two rounds of revisions to this manuscript. According to your valuable comments and suggestions to the manuscript (after the first round of revision), we made careful revisions to the manuscript in this third round of revision.

Based on the method development and data investigation in the previous studies, PALASR/MODIS forest and MODIS evergreen forest cover maps showed improved performance in tracking the area, spatial distribution, and temporal changes of forest and evergreen forest cover in the Brazilian Amazon. Canopy height and canopy coverage are two critical variables in defining forest. However, the previous accuracy assessment of forest cover maps was mainly based on the canopy coverage without canopy height. In this data manuscript, we did additional accuracy assessment for PALSAR/MODIS forest and MODIS evergreen forest cover maps using the ICESat-1 canopy height and canopy coverage data product. The ICESat-1 data showed high accuracy of PALSAR/MODIS forest and MODIS evergreen forest cover maps in the Brazilian Amazon. We wrote this data manuscript to share the data and code of our forest and evergreen forest cover maps with the research community.

As you suggested, we added the error matrices (Table S1) based on the two reference data.

This manuscript is about data introduction, so we did not do much forest change analysis. We have shared the annual forest and evergreen forest cover maps. People can download these data products and investigate forest changes depending on their needs. We added three sites of localized visual comparisons of very high spatial resolution images, PALSAR/MODIS forest cover and MODIS evergreen forest cover maps (Fig. 8).

To match the 50-m PALSAR data, we resampled the 250-m MODIS NDVImax into 50-m spatial resolution using the nearest sampling approach. We added this data processing in the section "2.3. MODIS surface reflectance and vegetation indices".

**Reviewer #5**

General comments:

This study presented a datasets of forest map according to FAO's definition, which is of great importance to the sustainable development and biomass estimation in specific. From the results presented in this paper, the accuracy is of good performance, and it has a great contribution for large-scale forest mapping work. However, I recommend the authors to address the following issues before the manuscript can be considered to be published on ESSD.

Response: Thanks for your positive comments. We made careful revisions to this manuscript according to your comments and suggestions.

1. The description of method about forest mapping is not comprehensive enough, even was ignored. Please express this part fully. The detail of methodology is missing, which is not acceptable without further modification regarding the scope of ESSD journal. It is not easy for readers to know how they produced the datasets without further knowledge from the cited paper.

Response: We added more details about the method description of forest mapping. Please see section "2.6. Algorithm and data of annual PALSAR/MODIS forest cover maps during 2007-2010".

"Electromagnetic wave of PALSAR can penetrate the tree canopy and interact with the tree trunks and branches. Forests have higher volume backscatter signals in HH and HV compared to croplands, grasslands, and water bodies. Thus, PALSAR data are sensitive to forest structure and biomass. However, PALSAR data can be affected by local incidence angle and soil moisture as PALSAR data is acquired at a different date each year. We calculated the acquisition date (Fig. 2), the local incidence angle (Fig. S1), and HH and HV gamma-naught values for each year and their standard deviations (Fig. S2) during 2007-2010 in the Brazilian Amazon. PALSAR HH and HV data were mainly acquired in the dry season (from June to October) and the local incidence angle is stable. About 90% of the area has standard deviation values of less than 1 dB for PALSAR HH and HV data. PALSAR data have advantages in identifying and mapping the spatial and temporal changes of forests in the tropics with frequent clouds compared to optical satellite remote sensing. Using the FAO's forest definition as the reference, we developed a robust decision tree algorithm to identify and generate forest cover maps by ALOS PALSAR data: $-15 \leq HV \leq -9$, $3 \leq Difference \leq 7$, and $0.35 \leq Ratio \leq 0.75$, based on the forest and non-forest training samples (Qin et al., 2016a; Qin et al., 2015; Qin et al., 2017). Several land cover types (e.g., rocks and buildings) had high backscatter values of HH and HV, which were often confused with the forests when only HH and HV data were used. These land cover types usually have low vegetation coverage with NDVImax < 0.5 (Qin et al., 2016a; Qin et al., 2015; Qin et al., 2017). To reduce the commission errors from these land cover types, we combined both PALSAR and NDVImax from MOD13Q1 to produce annual forest cover maps (namely PALSAR/MODIS) at 50-m spatial resolution in the Brazilian Amazon during 2007-2010 using these threshold values: $-15 \leq HV \leq -9$, $3 \leq Difference \leq 7$, $0.35 \leq Ratio \leq 0.75$, and NDVImax $\geq$ 0.5 (Qin et al., 2016a; Qin et al., 2017). We also carried out a three-year temporal consistency filter to reduce the effects of noise (Qin et al., 2016a; Qin et al., 2017)."

2. The availability of existing reference datasets needs to be further clarified, especially in the accuracy comparison part.

Response: We added the description of the availability of existing reference datasets in the revised manuscript (3.1. Annual PALSAR/MODIS forest cover maps during 2007-2010) as "The

Global Land Cover Validation Reference Dataset was produced from very high spatial resolution commercial remote sensing data acquired around 2010 and is freely available at the https://web.archive.org/web/20161209205946/https:/landcover.usgs.gov/glc/SitesDescriptionAnd Downloads.php. The TREES-3 dataset was produced from Landsat images and is freely available at https://forobs.jrc.ec.europa.eu/trees3/data."

3. For the statistical accuracy analysis, the authors used 5km as the pixels' resolution. Why not conduct such analysis on a higher resolution? The resolution of the cartographic results in this paper is already very low (e.g., 50 m, 250 m, 500 m). Using a lower resolution to is actually much more friendly to a better accuracy, but it's fake. Please reconduct the uncertainty analysis to make it more convincible like previous studies [1-4].

Response: Thanks for sharing these four previous studies. We added the reason that we selected 5-km spatial resolution as an optimal scale to aggregate the 50-m SAR and 500-m MODIS images in the second round of revision: "For the spatial comparison, to avoid the bias caused by different spatial resolutions, we aggregated the 50-m annual PALSAR/MODIS forest cover maps and 500-m (463-m) MODIS evergreen forest cover maps into 5-km pixels and calculated their average forest area fraction values within individual 5-km pixels.". Please see section "2.8. Spatial and statistical analysis".

We added three sites of zoom-in windows of very high spatial resolution images, PALSAR/MODIS forest cover maps, and MODIS evergreen forest cover maps using these previous studies as the reference.

4. Did the forest map from 2007 to 2010 (Fig. 5) also combine PALSAR data and MODIS? Please clarify it, because the "or" in the caption is not consistent with the description in the text. For 2007-2021 evergreen forest mapping, only MODIS data were used only, which is clear. Similar mistakes are quite often in the manuscript, such as Sections 3.1 and 3.2. Please recheck the manuscript thoroughly.

Response: The PALSAR/MODIS forest cover maps from 2007 to 2010 in Fig. 5 (in the manuscript after the first round of revision) are generated based on PALSAR and MODIS images. As we moved some figures from the main text to supplementary materials in the second round of revision, this Fig. 5 became Fig. 3. In this third round of revision, we highlighted the meaning of the "PALSAR/MODIS forest cover maps" in the section "2.6. Algorithm and data of annual PALSAR/MODIS forest cover maps during 2007-2010".

5. Please add zoomed images (e.g., optical images) for details analysis to prove your results is corrected. Please refer to the existing research analysis [1-4].

Response: As you suggested, we added the zoomed very high spatial resolution images, PALSAR/MODIS forest cover, and MODIS evergreen forest cover maps at three sites (Fig. 8).

6. Section 3.4 should be placed in the introduction instead of "Results and discussion" since section 3 described the advantages satellite lidar data for result assessment.

Response: We made this change in the second round of revision.

7. The comprehensive discussion of the results is inadequate in the manuscript, such as the possible reasons for misclassifications.

Response: We added one section in the revised manuscript.

"3.3. Uncertainties in the accuracy assessment of PALSAR/MODIS forest cover maps and MODIS evergreen forest cover maps

The Global Land Cover Validation Reference and TREES-3 land cover datasets were produced from optical remote sensing images, which were sensitive to canopy coverage instead of canopy height. Thus, our PALSAR/MODIS forest cover maps had high user's and producer's accuracy for forest cover type, while the non-forest cover type had relatively low user's or producer's accuracy (Table S1), which may be attributed to the uncertainties in the reference maps. Different from the optical remote sensing image, the ICESat-1 data used in this study had not only the maximum canopy coverage data but also the maximum canopy height from 2003 to 2007 (Tang et al., 2019a). As the Brazilian Amazon had high annual primary forest loss rates of 17,654 km$^2$/yr in the 2000s (INPE, 2023), the maximum canopy height and canopy coverage of ICESat-1 data may not include the impacts of deforestation. Thus, ~94% of PALSAR/MODIS forest cover pixels and MODIS evergreen forest cover pixels meet the forest definition."

8. Generally speaking, the source data used in this paper are quite outdated. (e.g.,? This deficiency leads to a much lower significance of this study. Why not use more modern data like Sentinel, ICESat-2, ALOS-2? This should be answered seriously, as well as in the text.

Response: In our opinion, each data product has its own advantages and disadvantages in forest cover mapping. Frequent cloud cover is a major challenge for forest cover mapping in the Brazilian Amazon. Although the 8-day MOD09A1 data product has a spatial resolution of 500m, it is generated based on daily observations and has a high opportunity to get cloud-free observations. Besides, MOD09A1 has data collection since 2000, which could track long-term forest cover changes due to frequent policy and environmental changes in the Brazilian Amazon, especially the different phases of deforestation. The ICESat-1 canopy height and canopy coverage percentage data products were generated by Tang et al (2019, Remote Sensing of Environment). ICESat-2 has the canopy height data product but its canopy coverage data product was not calculated from LiDAR observations instead of Landsat vegetation coverage fraction. Sentinel has data observations for about 10 years. The combination of Sentinel and Landsat could provide land surface observations about every 3-4 days. ALOS PALSAR-2 and Global Ecosystem Dynamics Investigation (GEDI) canopy height and canopy coverage data products

have been updated recently. Combining Sentinel, ALOS PALSAR-2, Landsat, and GEDI to generate annual forest cover maps is on our to-do list. We added the relevant description in the section "2.3. MODIS surface reflectance and vegetation indices".

10. The topic of this study, which is "annual forest" and "evergreen forest", as we can see in the title and abstract. But as I know, and as we can know from the results, these two types of forests are almost the same in Amazon. So why separate them apart if the difference is not significant (see Section 3.3 and Fig. 9)? Or, what's the difference between them and how we can tell it from the results?

Response: To assess the accuracy and uncertainty of MODIS evergreen forest cover maps, we used multiple independent reference datasets at the site scales, including land cover maps at the 2-m spatial resolution, land cover maps at the 30-m spatial resolution, and ICESat-2 canopy height and canopy cover percentage data products. These site-level reference datasets are sampling datasets, which did not cover the whole Brazilian Amazon. The major forest type is the evergreen forest in the Brazilian Amazon. Thus, we used the PALSAR/MODIS forest cover maps, which had little impact from frequent clouds, as the reference to assess the uncertainty of MODIS evergreen forest cover maps.

Reference:

[1] Shimada, Itoh, Motooka, et al. New Global Forest/Non-Forest Maps from ALOS PALSAR Data (2007-2010) [J]. Remote Sensing of Environment, 2014, 2014,155(-): 13-31.DOI:10.1016/j.rse. 2014.04.014.

[2] Martone M, Rizzoli P, Wecklich C, et al. The global forest/non-forest map from TanDEM-X interferometric SAR data[J]. Remote Sensing of Environment, 2018, 205:352373.DOI:10.1016/j.rse.2017.12.002.

[3] Mazza A, Sica F, Rizzoli P, et al.TanDEM-X Forest Mapping using Convolutional Neural Networks[J].Remote Sensing, 2019, 11(2980).DOI:10.3390/rs11242980.

[4] Pulella A, Santos R A, Sica F, et al. Multi-Temporal Sentinel-1 Backscatter and Coherence for Rainforest Mapping[J]. Remote Sensing, 2020, 12(5):847-.DOI:10.3390/rs12050847.

Specific comments:

1. Line 262: 0.03×106 -> 3×104; Line 248: 0.75×106 -> 7.5×105; please check such representation.

Response: We used the multiplication step for increments of the base-10 exponent of three to match thousand, million etc.

2. Line 165, Fig.1(a) and (b): The legend should be included in the figure.

Response: We included the legend in Figure 1(a) and (b).

3. Line 199, Fig.4(c) and (d): The label is too crowded to read.

Response: We made Fig. 4 bigger to be clear to read. We moved this figure into supplementary materials in the second round of revision.

4. Line 269, Fig.7: add a legend, like Fig. 5.

Response: We revised it as you suggested.

---

## Author Response (AR3)

**Response to Editors' Comments**

**Comments by Editor: Nophea Sasaki**

Dear Authors,

Thank you for addressing the concerns raised by me and the reviewers. However, the title remains confusing.

Original Title: Annual maps of forest and evergreen forest in the Brazilian Amazon from analyses of PALSAR and MODIS images

Based on your response that "These PALSAR/MODIS forest cover maps could include evergreen, deciduous, and mixed forests in the two major biomes. Evergreen forest cover is the dominant forest cover type in the Brazilian Amazon," the title may mislead readers. I recommend considering one of these revised titles or something similar:
1. Annual Maps of Forest Cover in the Brazilian Amazon from Analyses of PALSAR and MODIS Images
2. Annual Maps of Evergreen and Other Forest Cover in the Brazilian Amazon from Analyses of PALSAR and MODIS Images
3. Annual Maps of Evergreen Forest Cover in the Brazilian Amazon from Analyses of PALSAR and MODIS Images

If you choose the third option, please elaborate on evergreen forests in the study area section, including their definition and the reason for focusing specifically on this type.

Response: Dear Dr. Sasaki, We appreciate your suggestions for the title. We choose your first suggestion for the title "Annual Maps of Forest Cover in the Brazilian Amazon from Analyses of PALSAR and MODIS Images".

**Comments by Editor: Polina Shvedko**

Please ensure that the colour schemes used in your maps and charts allow readers with colour vision deficiencies to correctly interpret your findings. Please check your figures using the Coblis – Color Blindness Simulator (https://www.color-blindness.com/coblis-color-blindness-simulator/) and revise the colour schemes accordingly.
Response: Dear Dr. Shvedko, We checked the figures in the manuscript and modified the symbols for each forest data product in Figure 9.